# Membrane type 1 matrix metalloproteinase promotes LDL receptor shedding and accelerates the development of atherosclerosis

Adekunle Alabi[1,6], Xiao-Dan Xia[1,2,6], Hong-Mei Gu[1], Faqi Wang[1], Shi-Jun Deng[1], Nana Yang[3], Ayinuer Adijiang[1], Donna N. Douglas[4], Norman M. Kneteman[4], Yazhuo Xue[5], Li Chen[5], Shucun Qin[5], Guiqing Wang[2] & Da-Wei Zhang [1✉]

Plasma low-density lipoprotein (LDL) is primarily cleared by LDL receptor (LDLR). LDLR can be proteolytically cleaved to release its soluble ectodomain (sLDLR) into extracellular milieu. However, the proteinase responsible for LDLR cleavage is unknown. Here we report that membrane type 1-matrix metalloproteinase (MT1-MMP) co-immunoprecipitates and co-localizes with LDLR and promotes LDLR cleavage. Plasma sLDLR and cholesterol levels are reduced while hepatic LDLR is increased in mice lacking hepatic MT1-MMP. Opposite effects are observed when MT1-MMP is overexpressed. MT1-MMP overexpression significantly increases atherosclerotic lesions, while MT1-MMP knockdown significantly reduces cholesteryl ester accumulation in the aortas of apolipoprotein E (apoE) knockout mice. Furthermore, sLDLR is associated with apoB and apoE-containing lipoproteins in mouse and human plasma. Plasma levels of sLDLR are significantly increased in subjects with high plasma LDL cholesterol levels. Thus, we demonstrate that MT1-MMP promotes ectodomain shedding of hepatic LDLR, thereby regulating plasma cholesterol levels and the development of atherosclerosis.

[1] The Department of Pediatrics and Group on the Molecular and Cell Biology of Lipids, Faculty of Medicine and Dentistry, University of Alberta, Edmonton, AB, Canada. [2] Department of Orthopedics, The Sixth Affiliated Hospital of Guangzhou Medical University, Qingyuan People's Hospital, Qingyuan, China. [3] Experimental Center for Medical Research, Weifang Medical University, Weifang, China. [4] Department of Surgery, Faculty of Medicine and Dentistry, University of Alberta, Edmonton, AB, Canada. [5] Institute of Atherosclerosis in Shandong First Medical University (Shandong Academy of Medical Sciences), Taian, China. [6] These authors contributed equally: Adekunle Alabi, Xiao-Dan Xia. ✉email: dzhang@ualberta.ca

Atherosclerotic cardiovascular disease is one of the main causes of morbidity and mortality in Western societies. Plasma levels of low-density lipoprotein cholesterol (LDL-C) are positively correlated with risk of atherosclerosis[1]. LDL receptor (LDLR) mediates LDL uptake and plays an essential role in removing plasma LDL-C. Upon LDL binding, LDLR is internalized via clathrin-coated pits and delivered to endosomes, where LDL is released from the receptor and delivered to the lysosome for degradation, LDLR is then recycled to the cell surface[2]. Mutations in LDLR cause familial hypercholesterolemia and increase risk for atherosclerosis and coronary heart disease[1].

LDLR is transcriptionally regulated by the sterol regulatory element-binding protein 2 (SREBP-2). Statins inhibit 3-Hydroxy-3-Methylglutaryl-CoA Reductase (HMGCR) and consequently activate the transcriptional activity of SREBP2, leading to increased LDLR expression[1]. Posttranslationally, proprotein convertase subtilisin/kexin 9 (PCSK9) binds to LDLR and redirects the receptor for lysosomal degradation[3–8], while the inducible degrader of LDLR (IDOL) reduces LDLR levels via the polyubiquitination and lysosomal degradation pathway[9]. In addition, the ectodomain of LDLR can be cleaved by proteases with the released extracellular domain detected in cell culture media and in human plasma as a soluble form (sLDLR)[10–13]. Serum levels of sLDLR are positively correlated with plasma LDL-C levels[11,14]. It has been shown that ectodomain cleavage of LDLR is inhibited by metalloproteinase inhibitors[10,15]. Knockdown of a disintegrin and metalloproteinase 17 (ADAM17), a metalloproteinase responsible for the shedding of many transmembrane proteins, however, has only a marginal effect on ectodomain cleavage of LDLR in HepG2 cells[15]. Thus, the metalloproteinase(s) responsible for the bulk of LDLR shedding is unknown.

Membrane type-I matrix metalloproteinase (MT1-MMP/MMP14), a $Zn^{2+}$-dependent endopeptidase, belongs to a six-member family of membrane-type MMPs that includes four transmembrane type MMPs (MT1-, MT2-, MT3-, MT5-MMP), and two glycosyl phosphatidylinositol-anchored membrane-associated MMPs (MT4-, MT6-MMP)[16]. MT1-MMP is widely expressed in various tissues and cell types and plays key roles in both physiological processes and disease progression through remodelling of extracellular matrix and pericellular proteolysis[17–19]. MT1-MMP has been reported to cleave transmembrane proteins such as death receptor-6, neuropilin-1, and LDLR-related protein 1 (LRP-1) in breast cancer cells[20,21]. Cleavage of LDLR is enhanced by 4β-phorbol 12-myristate 13-acetate that can induce trafficking of MT1-MMP to the cell surface and enhance its cleavage function[10]. The exact role of MT1-MMP in LDLR shedding, however, is unclear. Here, we found that reducing MT1-MMP expression increased LDLR levels in cultured cells. Furthermore, specific knockout of MT1-MMP in mouse hepatocytes increased hepatic LDLR levels and reduced plasma levels of lipoprotein cholesterol. An opposite phenotype was observed when the wild-type MT1-MMP, but not the enzymatically dead mutant E240A, was overexpressed in cultured cells and mice. Consequently, overexpression of MT1-MMP significantly increased atherosclerotic lesion area in aortic sinus, while knockdown of MT1-MMP reduced cholesterol accumulation in the aortas in apolipoprotein E knockout (apoE$^{-/-}$) mice. Mechanistically, we found that MT1-MMP directly associated with LDLR and promoted its ectodomain cleavage. Taken together, these findings demonstrate that hepatic LDLR ectodomain is shed by MT1-MMP and that MT1-MMP regulates plasma LDL-C metabolism and the development of atherosclerosis.

## Results

**MT1-MMP regulates LDLR expression.** MT1-MMP activates pro-MMP2[17,22]. We have previously reported that active MMP2 can cleave PCSK9[23]. Thus, we initially assumed that inhibition of MT1-MMP could reduce the amount of active MMP2 and suppress MMP2-induced cleavage of PCSK9, thereby enhancing PCSK9-promoted LDLR degradation and consequently decreasing LDLR levels. To test this hypothesis, we knocked down expression of MT1-MMP in human hepatoma-derived Huh7.5 cells. The two MT1-MMP siRNAs that targeted different regions in the *MT1-MMP* mRNA efficiently reduced the levels of MT1-MMP but not MT2-MMP (the closest family member to MT1-MMP) (Fig. 1a). Addition of PCSK9 reduced cellular LDLR levels in scrambled siRNA-transfected cells, as well as in cells transfected with MT1-MMP siRNA (Fig.1b). We then co-transfected Huh7.5 cells with MT1-siRNA and plasmid containing PCSK9 cDNA and found that overexpression of PCSK9 efficiently stimulated LDLR degradation in cells transfected with either scrambled or MT1-MMP siRNA (Supplementary Fig. 1a). Thus, knockdown of MT1-MMP did not affect PCSK9-promoted LDLR degradation. Surprisingly, LDLR levels were markedly increased in MT1-MMP knockdown cells in the absence of PCSK9 (Fig. 1b, lanes 3 and 5 vs. 1). We noticed that knockdown of MT1-MMP appeared not to markedly affect the levels of LDLR in the presence of exogenous PCSK9 (Fig. 1b, lanes 4 and 6 vs. 2). It is of note that the experiment was performed in the presence of excess PCSK9 and under a non-physiological condition. First, Huh7.5 cells express endogenous PCSK9. Second, the cells were incubated in medium containing 5% NCLPPS that is known to increase endogenous PCSK9 expression and enhance PCSK9-promoted LDLR degradation. Third, the cells were supplied with additional 2 µg/ml of recombinant human PCSK9. Thus, it was likely that PCSK9-promoted LDLR degradation became overwhelming under this condition. To further confirm the impact of MT1-MMP on LDLR expression, we knocked down MT1-MMP expression in another human hepatoma-derived cell line (HepG2) and found that LDLR levels were significantly increased in MT1-MMP siRNA-transfected cells, whereas the levels of MT2-MMP, LRP-1 and transferrin receptor were comparable in cells transfected with scrambled or MT1-MMP siRNA (Fig. 1c, lanes 2 and 3 vs. 1). Similar results were observed in mouse hepatocytes, Hepa1c1c7 (Supplementary Fig. 1b). Next, we overexpressed HA-tagged MT1-MMP in Huh7.5 cells and observed that MT1-MMP reduced cellular LDLR levels in a dose-dependent manner (Fig. 1d). Overexpression of MT1-MMP, however, had no significant effect on the levels of endogenous PCSK9 (Supplementary Fig. 1c, lanes 4–6 vs. 1–3). LDLR resides on plasma membrane, where it binds to and mediates LDL internalization. Thus, cell surface proteins in Huh7.5 cells were assessed using biotinylation. As shown in Fig. 1e, expression of MT1-MMP in both whole-cell lysate and the cell surface fraction was reduced by its siRNA. Conversely, expression of LDLR was increased in whole-cell lysate (lane 2 vs. 1) and the surface fraction (lane 4 vs. 3) in MT1-MMP-knockdown cells. Calnexin (an ER protein) was undetectable in the surface fraction. Analysis of basal LDL uptake by these cells demonstrated a role for MT1-MMP in this process, with knockdown of MT1-MMP showing significantly increased cellular LDL uptake (Fig. 1f). Together, these findings demonstrate that MT1-MMP regulates LDLR expression and LDL uptake in cultured hepatocytes.

**MT1-MMP promotes ectodomain cleavage of LDLR.** We then examined what mechanisms mediated the effect of MT1-MMP on LDLR expression. Ilomastat (GM6001), a collagen-based peptidomimetic hydroxamate that can inhibit MT1-MMP activity[24],

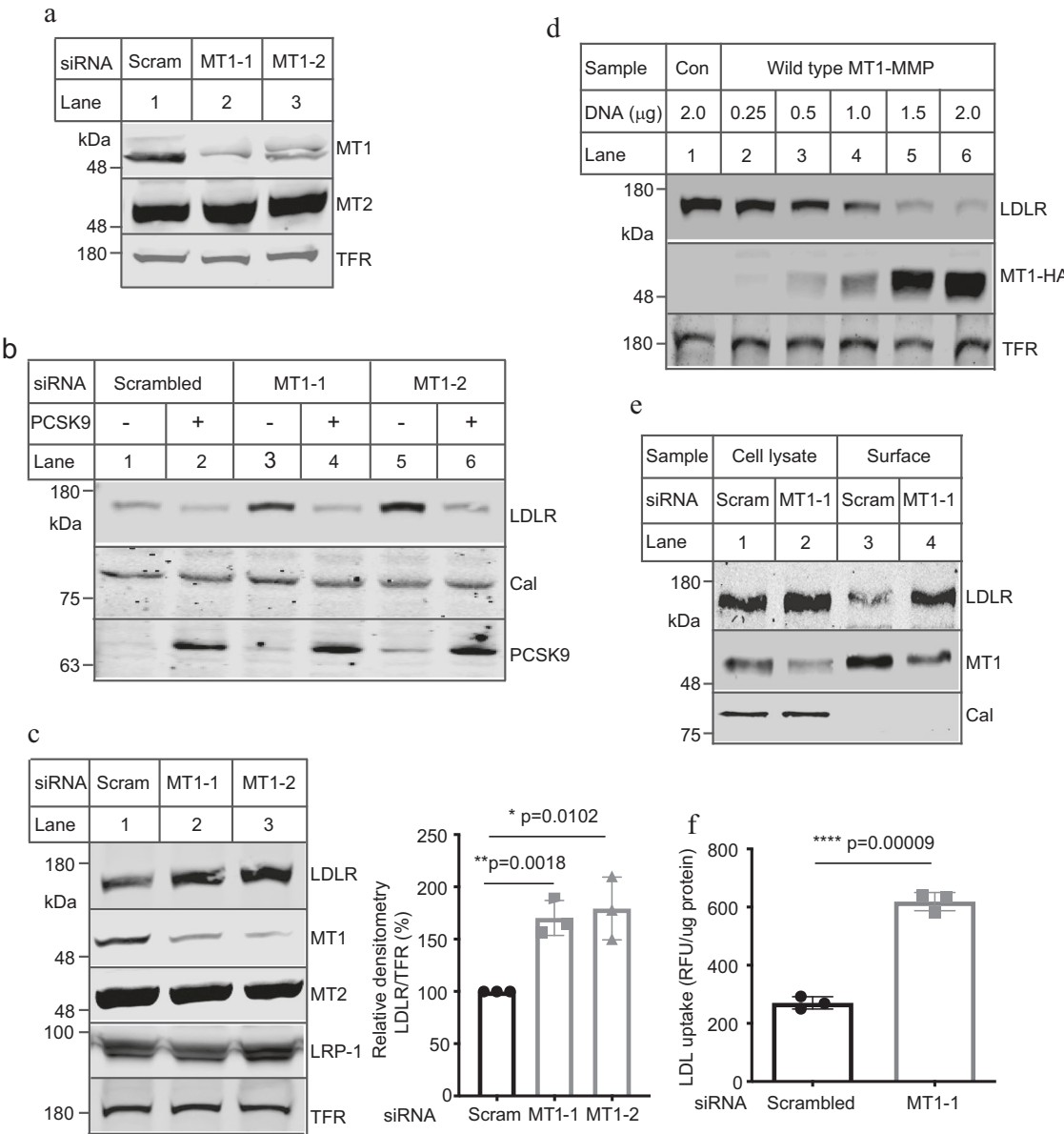

**Fig. 1 MT1-MMP-mediated LDLR degradation. a** Knockdown of MT1-MMP expression. Whole-cell lysate from Huh7.5 cells transfected with scrambled (Scram) or one of the two different MT1-MMP siRNAs (MT1-1, MT1-2) was applied to immunoblotting. TFR, transferrin receptor. **b** Effect of MT1-MMP knockdown on PCSK9-promoted LDLR degradation. Huh7.5 cells transfected with scrambled or MT1-MMP siRNA were incubated with or without PCSK9 (2 μg/ml). Whole-cell lysate was applied to western blot with antibodies indicated. **c** Effect of MT1-MMP knockdown in HepG2 cells. The cells were transfected with scrambled (Scram) or MT1-MMP siRNAs (MT1-1, MT1-2) for 48 h. Same amount of whole-cell lysate was applied to immunoblotting. The images showed representative protein levels. Similar results were obtained from three independent experiments. The relative densitometry was the ratio of the densitometry of LDLR to that of transferrin receptor (TFR) at the same condition ($n = 3$ independent experiments). The percentage of relative densitometry was the ratio of the relative densitometry of LDLR at different treatments to that of LDLR at the control condition, which was defined as 100%. **d** Effect of MT1-MMP overexpression on LDLR. Whole-cell lysate was isolated from Huh7.5 cells transfected with empty plasmid (Con) or different amount of plasmid carrying MT1-MMP cDNA, and then subjected to immunoblotting. **e** Biotinylation of cell surface proteins. Huh7.5 cells transfected with scrambled (Scram) or MT1-MMP siRNA (MT1-1) were biotinylated. Same amount of total proteins in whole-cell lysate was applied to NeutrAvidin beads to pull down cell surface proteins, followed by immunoblotting. Cal, calnexin. **f** LDL uptake ($n = 3$ independent experiments). Huh7.5 cells transfected with scrambled or MT1-MMP siRNA (MT1-1) were labeled with DiI-LDL in the absence and presence of unlabeled LDL. Fluorescence intensity (RFU) was measured to calculate specific binding. Similar results were obtained from at least three independent experiments. Student's *t* test (two-sided) was carried out to determine the significant differences between groups (**c** and **f**). The significance was defined as $p < 0.05$. Values of all data were mean ± SD. Source data are provided as a Source Data file.

did not alter expression of MT1-MMP but increased endogenous LDLR levels in Huh 7.5 cells (Supplementary Fig. 1d). Leupeptin (an inhibitor of serine and cysteine proteases) and pepstatin (an inhibitor of aspartyl proteases), however, had no effect on LDLR levels (Supplementary Fig. 1e). These findings were consistent with previous reports that LDLR cleavage depends on MMPs[10]. Moreover, MT1-MMP could efficiently promote LDLR degradation in the absence and presence of MG132, a proteasome inhibitor, or chloroquine, a lysosome inhibitor (Fig. 2a), indicating a negligible role of the two pathways in MT1-MMP's action on the

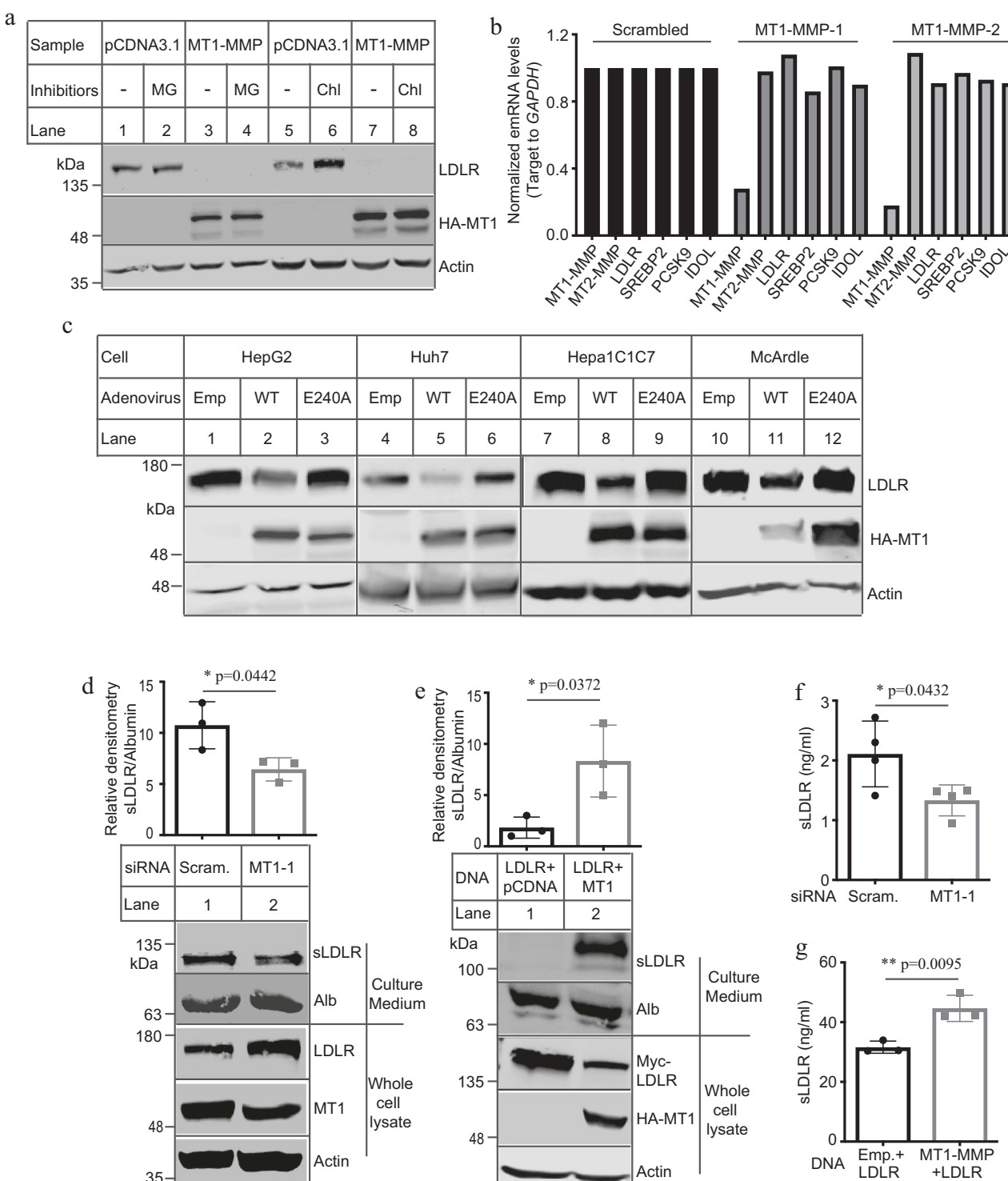

receptor. qRT-PCR data showed that both MT1-MMP siRNAs efficiently reduced mRNA levels of *MT1-MMP* but not that of *MT2-MMP*, *LDLR*, *SREBP-2*, *PCSK9*, or *IDOL* (Fig. 2b), indicating that MT1-MMP had no dramatic effect on transcription of *LDLR*. We then generated a catalytically inert mutant MT1-MMP by replacing Glu at position 240 (E240) with Ala (MT1-E240A) since E240 is critical for MT1-MMP proteolytic activity[25]. As shown in Fig. 2c, overexpression of the wild-type MT1-MMP but not MT1-E240A markedly reduced LDLR levels in HepG2, Huh7.5, Hepa1c1c7 and rat hepatoma-derived McArdle cells,

indicating the requirement of the proteolytic activity of MT1-MMP for its effect on LDLR expression. We then collected culture medium from Huh7.5 cells transfected with scrambled or MT1-MMP siRNA to measure cleaved ectodomain of LDLR. As shown in Fig. 2d, knockdown of MT1-MMP significantly reduced the intensity of an LDLR band with a molecular weight of ~100–120 kDa in culture medium. Concomitantly, LDLR abundance in whole-cell lysate was increased in MT1-MMP knockdown cells (Fig. 2d, lane 2 vs. 1). On the other hand, co-expression of N-terminal Myc-tagged LDLR and HA-tagged

**Fig. 2 MT1-MMP-mediated LDLR cleavage. a** Inhibitors treatment. Huh7.5 cells transfected with empty pCDNA3.1 or HA-tagged MT1-MMP-pCDNA3.1were incubated with MG132 (MG) or chloroquine (Chloro). Whole-cell lysate was subjected to immunoblotting. HA-tagged MT1-MMP was detected by an anti-HA antibody. **b** MT1-MMP knockdown. qRT-PCR of Huh7.5 cells transfected with scrambled or MT1-MMP siRNAs. The relative mRNA levels were the ratio of the mRNA levels of the target genes to that of *GAPDH* at the same condition. The fold-change of the relative mRNA levels of target gene expression in MT1-MMP siRNA treated groups was determined in comparison with that in the control group that was defined as 1. **c** MT1-MMP overexpression. Cells as indicated were infected with either empty (Emp), the wild-type (WT), or mutant E240A (E240A) MT1-MMP adenoviruses. Whole-cell lysate was subjected to immunoblotting. **d** LDLR cleavage ($n = 3$ independent experiments). Huh7.5 cells transfected with scrambled (Scram) or MT1-MMP siRNA (MT1-1) were cultured in DMEM only medium for 16 h. Whole-cell lysate (bottom) and concentrated media (top) were subjected to immunoblotting. MT1: MT1-MMP; Alb, albumin; TFR, transferrin receptor. **e** Ectodomain cleavage ($n = 3$ independent experiments). HEK293 cells were transfected with plasmid indicated. After 36 h, culture medium was changed to DMEM containing 0.5% FBS for overnight. Whole-cell lysate (bottom) and culture medium (top) were subjected to immunoblotting. The bottom figures in panels **d** and **e** showed representative protein levels. The relative densitometry was the ratio of the densitometry of sLDLR to that of albumin in culture medium in the same sample. Soluble LDLR. Culture medium was collected from Huh7.5 cells (**f**) ($n = 4$ independent experiments) or HEK293 cells (**g**) ($n = 3$ independent experiments) and subjected to measurement of sLDLR using ELISA. Similar results were obtained from at least three independent experiments. Student's *t* test (two-sided) was carried out to determine the significant differences between groups (**d**–**g**). The significance was defined as $p < 0.05$. Values of all data were mean ± SD. Source data are provided as a Source Data file.

MT1-MMP in HEK293 cells significantly reduced LDLR levels in whole-cell lysate but increased cleaved ectodomain of LDLR in culture medium that was detected by an anti-Myc antibody (Fig. 2e, lane 2 vs. 1). The size of the cleaved extracellular domain of LDLR was consistent with the size of the soluble ectodomain of LDLR reported in previous studies[10,13]. We also measured sLDLR levels in culture medium using ELISA and found that knockdown of MT1-MMP significantly reduced (Fig. 2f), while co-expression of MT1-MMP and LDLR significantly increased the levels of sLDLR in culture medium (Fig. 2g). Thus, MT1-MMP can proteolytically cleave LDLR.

Our next experiments were to investigate the interaction between MT1-MMP and LDLR. As shown in Fig. 3a, LDLR was immunoprecipitated from whole-cell lysate isolated from HepG2 cells by its specific monoclonal antibody, but not a monoclonal anti-Myc antibody. MT1-MMP was present only in the LDLR-immunoprecipitated sample (lane 1). Transferrin receptor was not detectable in the immunoprecipitated pellets. A reciprocal immunoprecipitation using an anti-MT1-MMP antibody to pull down MT1-MMP revealed that LDLR co-immunoprecipitated with MT1-MMP but not the anti-Myc antibody (Fig. 3b, lane 1 vs. 2). Similarly, immunoprecipitation of MT1-MMP from whole-cell lysate of Huh7.5 cells pulled down LDLR (Fig. 3c, lane 2 vs. 1) and vice versa (Fig. 3d, lane 1 vs. 2). We also immunoprecipitated LDLR from Huh7.5 cells overexpressing the HA-tagged wild-type or E240A mutant MT1-MMP. Consistently, LDLR was only pulled down by its specific antibody but not the anti-Myc antibody or protein-G beads only (Supplementary Fig. 1f, lanes 3, 6 and 9). We found that HA-tagged E240A was co-immunoprecipitated with LDLR (lane 9). A long exposure of the membrane showed that a small amount of the wild-type MT1-MMP was present in the LDLR-immunoprecipitated sample, but not in others (lanes 6 vs. 4 and 5). It was likely that overexpressed wild-type MT1-MMP cleaved LDLR when they were co-localized since LDLR levels in the wild-type MT1-MMP-expressing cells were much lower than that in the control and MT1-E240A-expressing cells (Supplementary Fig. 1f, lane 11 vs. 10 and 12). To further confirm these findings, we performed confocal microscopy and found that a majority of endogenous MT1-MMP (green) and LDLR (red) could be detected on the cell periphery and co-localized in Huh7.5 cells (Fig. 3e, yellow in Merged panel). Similarly, HA-tagged MT1-E240A mainly resided on the cell surface and colocalized with endogenous LDLR when expressed in HepG2 cells (Supplementary Fig. 1g). Together, these findings indicate that MT1-MMP associates with LDLR.

We next determined the effects of MT1-MMP on LDLR expression in human primary hepatocytes that are more representative of the functions of human liver than immortalized human hepatoma-derived cell lines, such as HepG2 and Huh7.5. As shown in Fig. 3f, MT1-MMP siRNA reduced mRNA levels of MT1-MMP but not that of LDLR. Conversely, LDLR protein levels were markedly increased in MT1-MMP knockdown cells (Fig. 3g) but reduced in MT1-MMP-overexpressing cells (Fig. 3h). Consistently, sLDLR in the culture medium was reduced when expression of MT1-MMP was silenced (Fig. 3i) but increased when MT1-MMP was overexpressed (Fig. 3j). Collectively, these findings demonstrate that MT1-MMP promotes ectodomain cleavage of LDLR.

**MT1-MMP promotes LDLR cleavage in vivo.** We further sought to understand the regulatory role of MT1-MMP in LDLR expression in vivo. Considering that MT1-MMP null mice die at 3–4 weeks[17,18] and that the catabolism of plasma LDL-C is mainly mediated via hepatic LDLR, we generated MT1-MMP liver-specific knockout mice (*MT1*LKO) through crossing *MT1*Flox mice with Alb-Cre mice that express Cre recombinase under the control of a hepatocyte-specific albumin promoter (Supplementary Fig. 2a). The *MT1*Flox mice, in which exons 2 and 4 of the *MT1-MMP* gene (containing prodomain and catalytic domain) are flanked by LoxP sites, have been successfully used to generate knockout of MT1-MMP in monocytes/macrophages and epidermis[26,27]. The expression of MT1-MMP in primary hepatocytes isolated from mouse livers was markedly reduced in *MT1*LKO mice (Fig. 4a). Genotyping results also revealed that the *MT1-MMP* gene was essentially undetectable in the liver of *MT1*LKO but not *MT1*Flox mice (Supplementary Fig. 2b) or other tissues isolated from *MT1*LKO mice (Supplementary Fig. 2c). mRNA levels of *MT1-MMP* were also drastically reduced in the liver but not in other tissues of *MT1*LKO mice (Fig. 4b). *MT1*LKO mice were active, fertile and indistinguishable from floxed littermates (Supplementary Fig. 2d). Body weight and plasma ALT activities were also comparable in *MT1*LKO and *MT1*Flox mice (Supplementary Fig. 2e, f). We also did not observe a significant difference in mRNA levels of liver *MT2-MMP* and *Aadam17* (Fig. 4c), indicating that the loss of hepatic MT1-MMP was not compensated by these metalloproteinases. *MT1*LKO mice, however, did not display a significant collagen difference in Mason's trichrome staining of liver sections (Supplementary Fig. 3a) and plasma active MMP2 (Supplementary Fig. 3b). Thus, knockout of MT1-MMP in hepatocytes did not cause obvious liver damage.

Consistent with our findings in cultured cells, the protein levels of LDLR but not LRP1 in liver homogenate were significantly increased (Fig. 4d), while the levels of plasma sLDLR were significantly reduced in *MT1*LKO mice (Fig. 4e). On the other

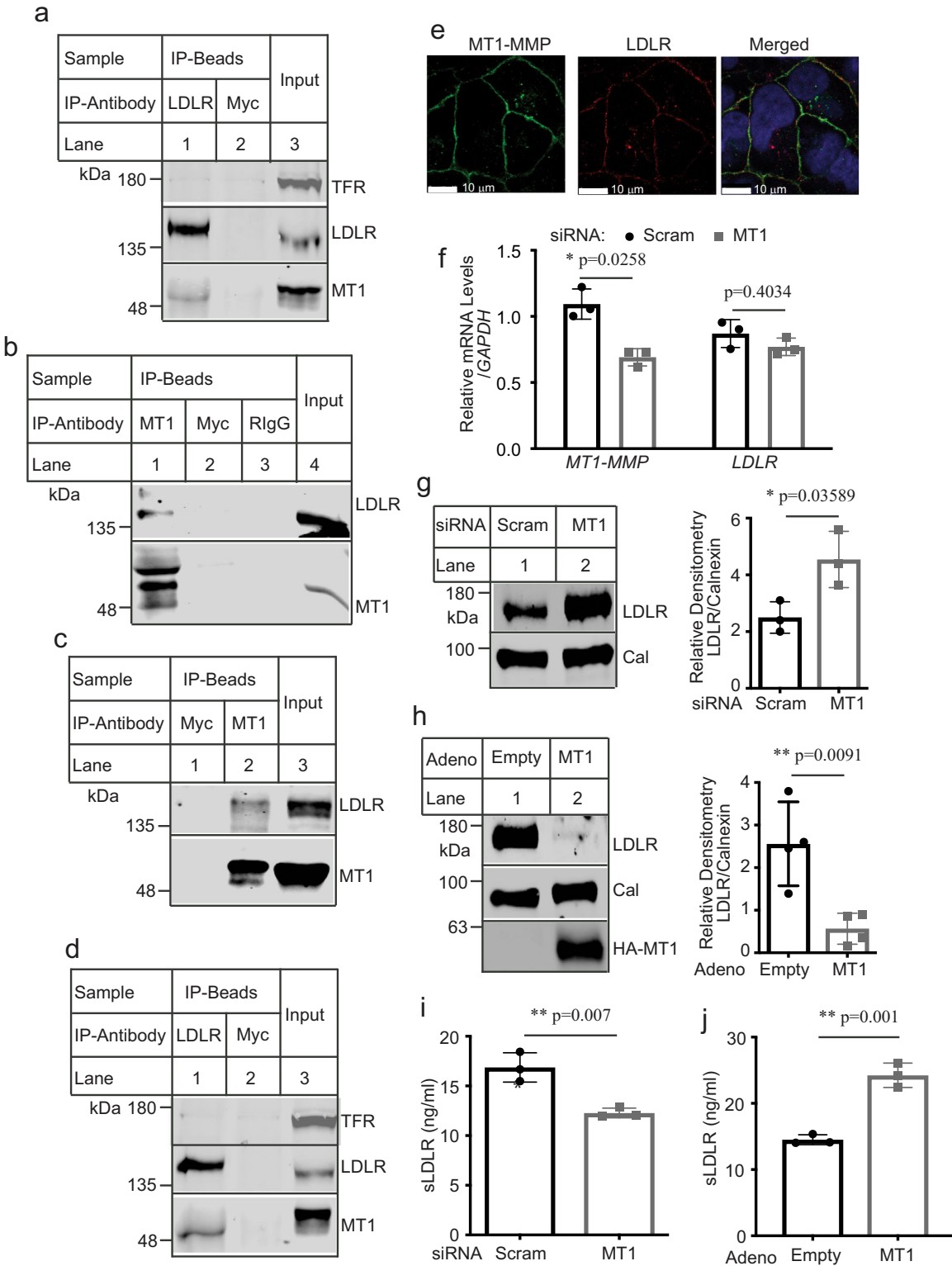

hand, the levels of PCSK9 in the liver homogenate and plasma were comparable in $MT1^{LKO}$ mice and $MT1^{Flox}$ mice (Supplementary Fig. 3c, d). We have previously reported that active MMP2 inhibited PCSK9-promoted LDLR degradation in hepa1c1c7 cells[23]. However, the levels of plasma pro- and active forms of MMP2 were not significantly altered in $MT1^{LKO}$ mice (Supplementary Fig. 3b), implying that MMP2 did not play an important role in the action of MT1-MMP on LDLR in mice. Furthermore, mRNA levels of *Ldlr*, *Srebp2*, *Hmgcr* and *Pcsk9* in

mouse liver were also not affected by hepatic deficiency of MT1-MMP (Fig. 4f). On the other hand, $MT1^{LKO}$ mice displayed a mild but significant reduction in plasma levels of total cholesterol (Fig. 4g). FPLC data showed that cholesterol levels in both LDL and HDL fractions were mildly reduced (Fig. 4h). Conversely, plasma TG levels were comparable in $MT1^{LKO}$ and $MT1^{Flox}$ mice (Supplementary Fig. 3e). Similarly, the lack of MT1-MMP in hepatocytes did not significantly affect the levels of liver TG or total cholesterol (Supplementary Fig. 3f, g). To confirm the

**Fig. 3 Effects of MT1-MMP on LDLR.** Immunoprecipitation of LDLR (**a** and **d**) and MT1-MMP (**b** and **c**). Whole-cell lysate from HepG2 (**a** and **b**) or Huh7.5 cells (**c** and **d**) was subjected to immunoprecipitation using protein G beads and a mouse anti-LDLR (LDLR) or anti-Myc (Myc) antibody (**a** and **d**), or protein G beads and a mouse anti-MT1-MMP antibody, a mouse anti-Myc (Myc) antibody or rabbit IgG (RIgG) (**b** and **c**). The immunoprecipitated proteins (IP-Beads) and whole-cell lysate (Input) were subjected to immunoblotting. Representative images were shown. Similar results were obtained from three independent experiments. **e** Confocal microscopy. Huh7.5 cells were fixed, permeabilized, and then incubated with a mouse anti-LDLR monoclonal and a rabbit anti-MT1-MMP monoclonal antibody. MT1-MMP: green; LDLR: red, DAPI: blue. An x-y optical section of the cells illustrates the cellular distribution of proteins (magnification: 100×). **f, g** MT1-MMP knockdown (n = 3 independent experiments). Primary human hepatocytes were transfected with scrambled (Scram) or MT1-MMP siRNA (MT1). The relative mRNA levels were the ratio of the mRNA levels of the target genes to that of *GAPDH* at the same condition (**f**). Same amount of total proteins in whole-cell lysate was subjected to immunoblotting. Cal: calnexin. Relative densitometry was the ratio of the densitometry of LDLR to that of calnexin at the same condition. **h** Overexpression of MT1-MMP (n = 4 independent experiments). Whole-cell lysate was prepared from primary human hepatocytes infected with empty or the wild-type MT1-MMP (MT1)-adenovirus and then applied to immunoblotting. sLDLR. Medium was collected from primary human hepatocytes treated with siRNA (**i**) or adenovirus (**j**) to measure sLDLR with ELISA. Similar results were obtained from at least three independent experiments. Student's *t* test (two-sided) was carried out to determine the significant differences between groups (**f–j**). The significance was defined as *p* < 0.05. Values of all data were mean ± SD. Source data are provided as a Source Data file.

specific contribution of MT1-MMP to LDLR and plasma cholesterol levels, we introduced human MT1-MMP into hepatocytes of $MT1^{LKO}$ via AAV under the control of a hepatocyte-specific TBG promoter. HA-tagged MT1-MMP was detected in the liver homogenate by a polyclonal anti-HA antibody (Fig. 4i). Reintroduction of MT1-MMP essentially eliminated the increase in liver LDLR (Fig. 4i) and the reduction in plasma levels of sLDLR (Fig. 4j) and total cholesterol (Fig. 4k) in $MT1^{LKO}$ mice. These findings indicate the important role of MT1-MMP in the regulation of liver LDLR levels.

We then overexpressed human MT1-MMP in the wild-type C57BL/6J mice using adenovirus under the control of a CMV promoter. As shown in Fig. 5a, overexpression of MT1-MMP significantly reduced LDLR levels in the liver. Consistently, plasma levels of HDL (Fig. 5b) and non-HDL cholesterol (Fig. 5c) were significantly increased. Mice on a chow diet normally have very low levels of plasma non-HDL-C. To further confirm our findings, we fed $MT1^{LKO}$ and $MT1^{Flox}$ mice with the Western-Type diet for 8 weeks. The protein levels of LDLR but not LRP1 in liver homogenate were significantly increased in $MT1^{LKO}$ mice (Fig. 5d). Masson's Trichrome and Oil Red-O staining of liver sections, however, displayed no significant difference between $MT1^{LKO}$ and $MT1^{Flox}$ mice (Fig. 5e and f). $MT1^{LKO}$ mice also did not display a significant difference in the levels of liver TG ($p =$ 0.1) and total cholesterol ($p = 0.073$) compared to $MT1^{Flox}$ mice (Supplementary Fig. 4a, b). The mRNA levels of *Ldlr* as well as other SREBP2 target genes (*Hmgcr* and *Pcsk9*) were also not altered by hepatic deficiency of MT1-MMP (Supplementary Fig. 4c). In addition, $MT1^{LKO}$ mice showed similar body weight gain (Supplementary Fig. 4d), H&E staining of liver sections (Supplementary Fig. 4e), plasma ALT activities (Supplementary Fig. 4f) and plasma levels of active MMP2 (Supplementary Fig. 4g, h) as $MT1^{Flox}$ mice. Conversely, hepatic knockout of MT1-MMP significantly and markedly reduced plasma levels of non-HDL and HDL cholesterol (Supplementary Fig. 5g, h). FPLC results revealed that cholesterol levels in all fractions including VLDL/chylomicron remnants, LDL and HDL were significantly reduced in $MT1^{LKO}$ mice (Fig. 5i). Thus, MT1-MMP regulates hepatic LDLR and plasma cholesterol levels in mice.

**sLDLR and plasma lipoproteins.** The molecular mass of the main form of sLDLR detected in our study was around 100 kDa that should consist of the entire ligand-binding domain of the receptor, indicating that sLDLR might retain the ability to bind LDLR ligands. Thus, we hypothesized that plasma sLDLR could bind to circulating apoB and apoE-containing lipoproteins. To test this possibility, we utilized gel filtration to determine the distribution of sLDLR in fasting mouse plasma. As shown in

Fig. 6a, sLDLR was eluted at fractions 8–17 with a wide peak at fractions 11–13 that were the fractions of LDL even though LDL only accounted for a very small portion of plasma cholesterol (Fig. 6b). There was a shoulder at fraction 15 that was partially overlapped with HDL. An additional small peak of sLDLR was present in the VLDL fractions 8–10. These could be caused by binding of sLDLR to apoE in VLDL and HDL particles. We also immunoprecipitated sLDLR from mouse plasma. The antibody efficiently pulled down sLDLR from plasma of the wild-type mice but not the $Ldlr^{-/-}$ mice (Fig. 6c), indicating the specificity of the antibody. Both apoB100 and apoB48, as well as apoE, were co-immunoprecipitated with sLDLR only from the wild-type mouse plasma (Fig. 6c, lane 2) even though the levels of apoB and apoE were much higher in $Ldlr^{-/-}$ mouse plasma (Fig. 6d, lane 1). Next, we analyzed human plasma using gel filtration chromatography. As shown in Fig. 6e, the majority of sLDLR were eluted at the fractions of the VLDL/chylomicron remnants and LDL even though VLDL/chylomicron remnants only accounted for a minor portion of plasma cholesterol (Fig. 6f). In addition, we noticed that, unlike mouse sLDLR, only a small peak of human sLDLR was eluted at the HDL fractions (Fig. 6e, f). We then measured sLDLR levels in purified human lipoproteins. As shown in Fig. 6g, VLDL exhibited the highest levels of sLDLR per μg proteins, followed by LDL and HDL. The levels of sLDLR in VLDL were ~35-fold more than that in LDL. The discrepancy between purified lipoproteins and plasma samples could be simply due to the relatively lower levels of VLDL present in human plasma. Taken together, these findings suggest that sLDLR associates with apoB and apoE-containing lipoproteins in plasma.

Next, we recruited 148 adult Chinese and measured their plasma levels of total cholesterol, LDL cholesterol, and sLDLR. There were 87 men (average age = 52.6), 46 women (average age = 53.7), and 15 individuals whose gender and age were undisclosed. The levels of sLDLR were comparable among the men, women and undisclosed group (Supplementary Fig. 5a). There was no significant correlation between plasma levels of sLDLR and ages (Supplementary Fig. 5b, $r = 0.047$, $p = 0.5925$). On the other hand, the correlation between sLDLR and plasma total cholesterol levels was statistically significant (Fig. 6h, $r = 0.41$, $p = 0.00000028$). Plasma LDL-C levels were also significantly correlated to sLDLR levels (Supplementary Fig. 5c, $r = 0.198$, $p = 0.0199$), but to a lesser extent than plasma total cholesterol levels. We also divided participants into three groups based on their plasma levels of total cholesterol or LDL cholesterol, group 1) the normal/desirable cholesterol levels (total cholesterol <5.2 mM and LDL cholesterol <3.4 mM, $N = 87$), group 2) the medium/borderline high plasma cholesterol levels (5.2 mM ≤ total cholesterol <6.1 mM or 3.4 mM ≤ LDL cholesterol < 4.1 mM, $N = 40$),

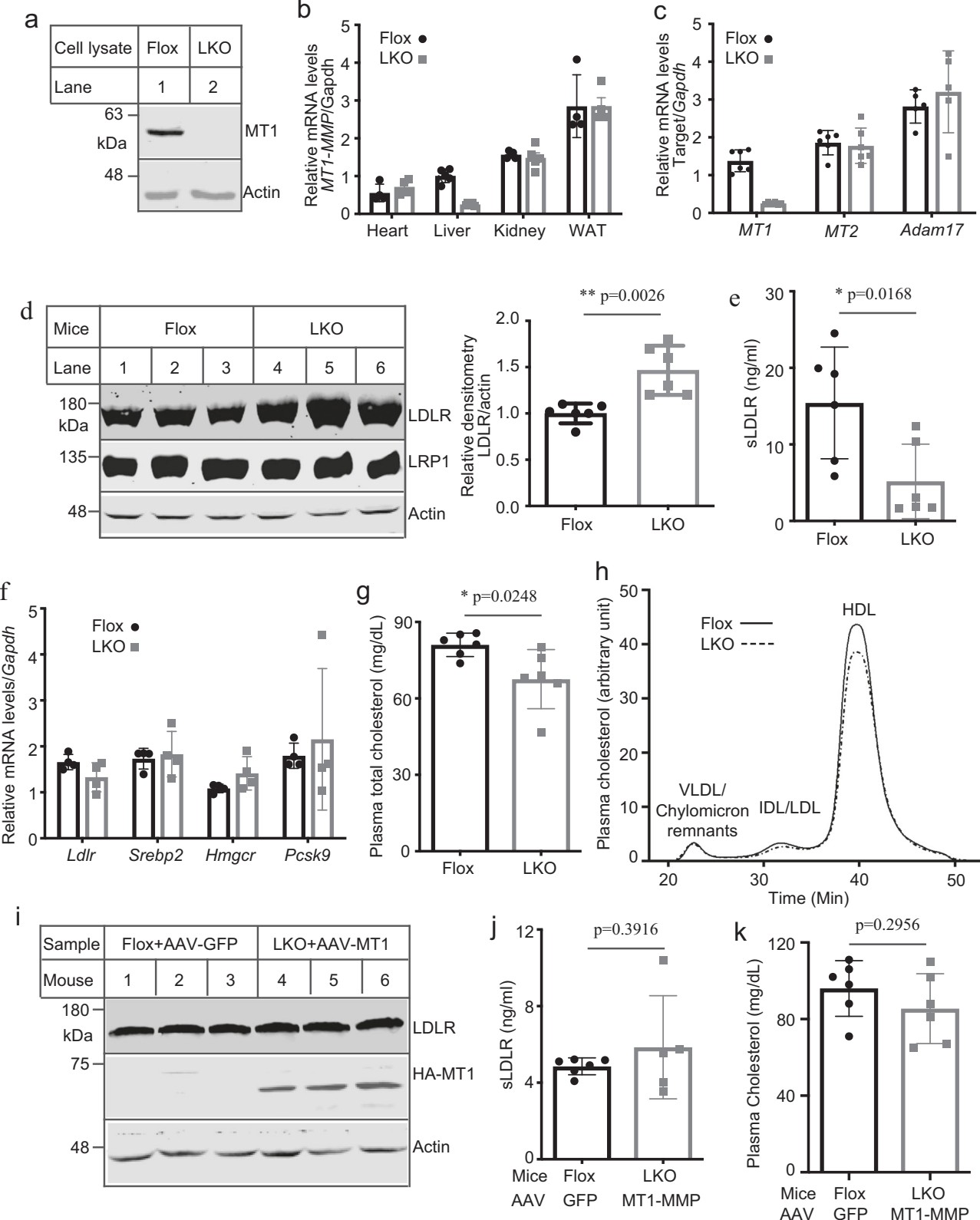

and group 3) the high plasma cholesterol levels (total cholesterol ≥ 6.1 mM or LDL cholesterol ≥ 4.1 mM, $N = 21$). The levels of sLDLR in the medium and high groups were significantly higher than that in the normal group, while there was no significant difference between the medium and high group (Fig. 6i). Given the critical role of PCSK9 in regulation of plasma LDL cholesterol and hepatic LDLR levels, we also measured circulating PCSK9 in

these subjects. As shown in Supplementary Fig. 5d, e, plasma levels of PCSK9 did not significantly associate with plasma LDL cholesterol in the whole group or women but did exhibit a positive correlation with plasma LDL cholesterol levels in men ($r = 0.2255$, $p = 0.0303$; Supplementary Fig. 5f). On the other hand, there was no significant association between plasma levels of PCSK9 and sLDLR in the whole group, women or men (Supplementary

**Fig. 4 Metabolic effects of *MT1*<sup>LKO</sup> mice. a** MT1-MMP deletion. Whole-cell lysate of primary mouse hepatocytes isolated from *MT1*<sup>Flox</sup> and *MT1*<sup>LKO</sup> mice was subjected to immunoblotting. Similar results were obtained from three independent experiments. **b, c** Relative expression of target genes. qRT-PCR measurement of mRNA levels of *MT1-MMP* in different tissues (**b**) (four or five mice per group) or *MT1-MMP, MT2-MMP,* and *Adam17* in the liver (five or six mice per group) (**c**). The relative mRNA levels were the ratio of the mRNA levels of the target genes from different tissues to that of *Gapdh* in the same tissue. WAT, white adipose tissue. **d** Liver LDLR levels (6 mice per group). Liver homogenate was subjected to immunoblotting. Relative densitometry was the ratio of the densitometry of LDLR of different mice to that of actin of the same mouse. Representative images were shown. Similar results were obtained from the other three mice. **e** Plasma sLDLR determined by ELISA (6 mice per group). **f** Relative mRNA levels (4 mice per group). **g** Plasma levels of total cholesterol (6 mice per group). **h** Lipid profile. Same amount of plasma from each mouse in the same group was pooled and applied to FPLC analysis of plasma cholesterol (6 mice per group). **i–k** Expression of human MT1-MMP. *MT1*<sup>Flox</sup> and *MT1*<sup>LKO</sup> mice were injected with AAVs encoding GFP or human MT1-MMP, respectively. Liver homogenate was applied to immunoblotting (6 mice per group). Representative images were shown. Similar results were obtained from the other three mice (**i**). Plasma samples were used to measure sLDLR levels (6 mice in the control group and 5 mice in MT1-MMP overexpression group) (**j**), and total cholesterol levels (6 mice per group) (**k**). Student's *t* test (two-sided) was carried out to determine the significant differences between groups (**b–g, j** and **k**). The significance was defined as *p* < 0.05. Values of all data were mean ± SD. Source data are provided as a Source Data file.

Fig. 5g–i). Together, these findings indicate that plasma levels of cholesterol but not PCSK9 are positively correlated to sLDLR levels.

**Effects of MT1-MMP on the development of atherosclerosis.** We then examined the effect of MT1-MMP on the development of atherosclerosis in mice. ApoE<sup>−/−</sup> mice (The Jackson Laboratory) were injected with empty AAV or AAV containing human MT1-MMP cDNA and then fed the Western Diet (TestDiet, 5TJN) for 8 weeks. Overexpression of MT1-MMP reduced LDLR levels in the liver (Supplementary Fig. 6a) and significantly increased plasma total cholesterol levels (791 mg/dL in the control, 896 mg/dL in MT1-MMP overexpressing mice, *p* = 0.0283; Supplementary Fig. 6b) and lesion area of aortic sinuses in apoE<sup>−/−</sup> mice (157.5 ± 21.03 μm<sup>2</sup> x 10<sup>3</sup> in the control group and 238.0 ± 21.23 μm<sup>2</sup> x 10<sup>3</sup> in MT1-MMP overexpressing mice, *p* = 0.0228; Fig. 6j). Next, we generated *MT1*<sup>Flox</sup>/apoE<sup>−/−</sup> mice through crossing *MT1*<sup>Flox</sup> mice with apoE<sup>−/−</sup> mice. Cre recombinase was then introduced into *MT1*<sup>Flox</sup>/apoE<sup>−/−</sup> mice via AAV under the control of a hepatocyte-specific TBG promoter (AAV-TBG-cre) to silence hepatic MT1-MMP expression. *MT1*<sup>Flox</sup>/apoE<sup>−/−</sup> mice injected with AAV-TBG-cre or a control AAV-GFP were then fed the Western Diet from Research Diets Inc. (D12079B) for 8 weeks. The mRNA levels of MT1-MMP were significantly reduced (Supplementary Fig. 6c), while hepatic LDLR levels were significantly increased in mice injected with AAV-TBG-Cre compared to the AAV-GFP injected mice (Supplementary *p* = 0.0385, Fig. 6d). However, knockdown of hepatic MT1-MMP did not significantly affect plasma total cholesterol levels (*p* = 0.2578, Fig. 6k) or lesion sizes of the aortic sinuses in *MT1*<sup>Flox</sup>/apoE<sup>−/−</sup> mice (Supplementary Fig. 6e, *p* = 0.6081). This was similar to a phenotype reported in the study of PCSK9 that knockout of PCSK9 in apoE<sup>−/−</sup> mice did not significantly affect plasma cholesterol levels or atherosclerotic plaque sizes despite of increased hepatic LDLR levels[28]. Denis et al. reported that knockout of PCSK9 reduced cholesteryl ester accumulated in the aortas of apoE<sup>−/−</sup> mice by ~39%, even though the sizes of the entire lesions were not significantly altered[28]. We employed the same approach and found a similar phenotype, MT1-MMP knockdown caused a significant reduction of ~33% in cholesteryl ester accumulation in the aorta of *MT1*<sup>Flox</sup>/apoE<sup>−/−</sup> mice (mean value: 7.046 and 4.706 μg of cholesteryl ester per aorta in AAV-GFP-injected and MT1-MMP knockdown mice, respectively, *p* = 0.0463 Fig. 6l). Together, these findings indicate that MT1-MMP promotes ectodomain shedding of LDLR and accelerates the development of atherosclerosis.

## Discussion

LDLR and its family members such as VLDLR, apoER and LRP1 undergo ectodomain shedding to release their soluble forms into the extracellular milieu such as cerebrospinal fluid and blood[12,29,30]. Recently, Girona et al reported that plasma levels of sLDLR were positively associated with non-HDL-C and small LDL numbers in FH children[31]. Consistently, two independent studies revealed a mild but significant correlation between serum concentrations of sLDLR and LDL-C in healthy adult Japanese and Canadian white population[11,14]. Here, we also found that sLDLR levels were positively correlated to plasma levels of total cholesterol and LDL cholesterol and significantly increased in subjects with high plasma levels of total or LDL cholesterol in adult Chinese. In addition, we observed a weak positive association between plasma levels of PCSK9 and LDL cholesterol in men but not in the whole group or women, consistent with a previous report[32]. Conversely, PCSK9 did not significantly correlate to sLDLR in men, women or the whole group. Circulating PCSK9 is mainly secreted from the liver but the underlying mechanism is unclear[33]. Loss-of-function PCSK9 mutations increase hepatic LDLR levels, which should render more receptors for MT1-MMP-mediated shedding and increase plasma sLDLR levels. Conversely, gain-of-function PCSK9 mutations reduce the amount of hepatic LDLR susceptible for shedding, which should reduce sLDLR in plasma. However, both loss-of-function and gain-of-function mutations can impair PCSK9 secretion and reduce its plasma levels[34–38]. Thus, a detailed analysis of genotype-phenotype association might be needed to explore the correlation between circulating PCSK9 and sLDLR. Nevertheless, these findings indicate that LDLR shedding plays an important role in lipid metabolism.

The proteinase responsible for LDLR ectodomain shedding, however, remains unknown. In the present study, we found that (1) MT1-MMP interacted with LDLR and knockdown of MT1-MMP expression in different cell types including human primary hepatocytes increased cellular LDLR levels, but decreased sLDLR levels in culture medium, (2) knockout of hepatic MT1-MMP in mice increased LDLR in the liver and reduced plasma levels of sLDLR, (3) overexpression of MT1-MMP reduced LDLR in cultured cells and mouse liver and increased sLDLR in culture medium and mouse plasma, but had no effect on PCSK9 expression, (4) knockdown of MT1-MMP had no effect on the mRNA levels of LDLR and other SREBP2 target genes including *PCSK9, SREBP2* and *HMGCR,* and (5) lysosomal inhibition with chloroquine that suppresses PCSK9- and IDOL-promoted LDLR degradation had no effect on MT1-MMP-induced reduction in LDLR. MT1-MMP knockdown also had no effects on the mRNA levels of *PCSK9* and *IDOL.* Taken

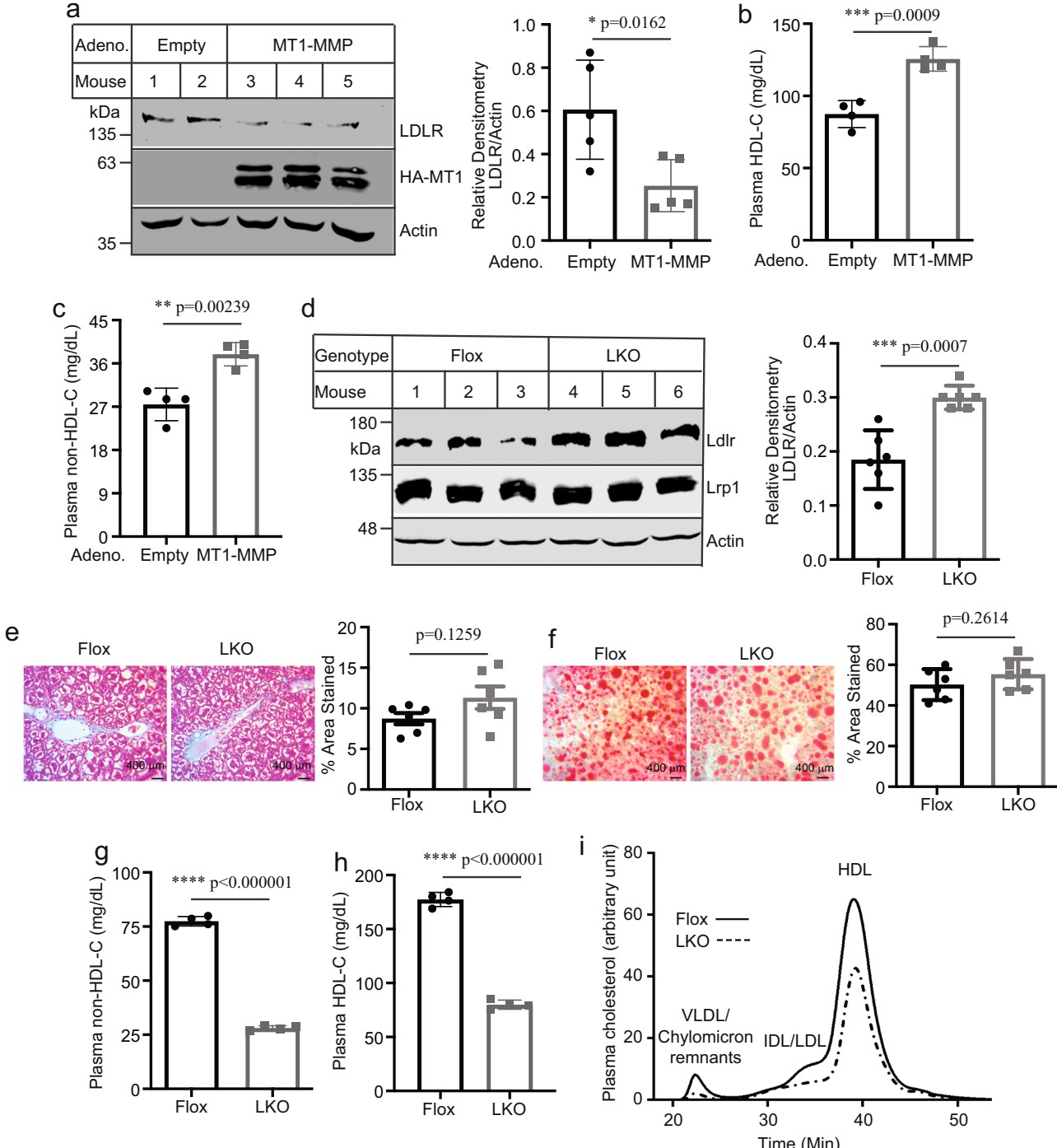

**Fig. 5 The impact of MT1-MMP. a** Overexpression of human MT1-MMP. Male C57BL/6J mice were injected with empty or the wild-type MT1-MMP adenoviruses (5 mice per group). Liver homogenate was subject to immunoblotting, followed by quantification of LDLR levels relative to actin in the same mouse. Representative images were shown. Similar results were obtained from other mice. Plasma levels of HDL (**b**) and non-HDL cholesterol (**c**) (4 mice per group). **d** Effect of the Western-type diet (6 mice per group). Mice were fed the Western-type diet for 8 weeks. Liver homogenate was subjected to immunoblotting. Relative densitometry of LDLR was determined as described. Actin was used as a loading control. Representative images were shown. Similar results were obtained from the other mice. Liver section staining. Representative figures and quantification (ImageJ 1.52S) of Masson's Trichrome (**e**) and oil Red-O staining (**f**) in cross-sections of the liver (6 mice per group). Similar results were obtained from the other mice (magnification: 400×). Plasma levels of cholesterol content in HDL (**g**) and non-HDL (**h**) (4 mice per group). **i** Lipid profile. Same amount of plasma from each mouse in the same group (2 female and 2 male mice per group) was pooled and applied to FPLC analysis of plasma cholesterol. Student's *t* test (two-sided) was carried out to determine the significant differences between groups (**a**–**h**). The significance was defined as *p* < 0.05. Values of all data were mean ± SD. Source data are provided as a Source Data file.

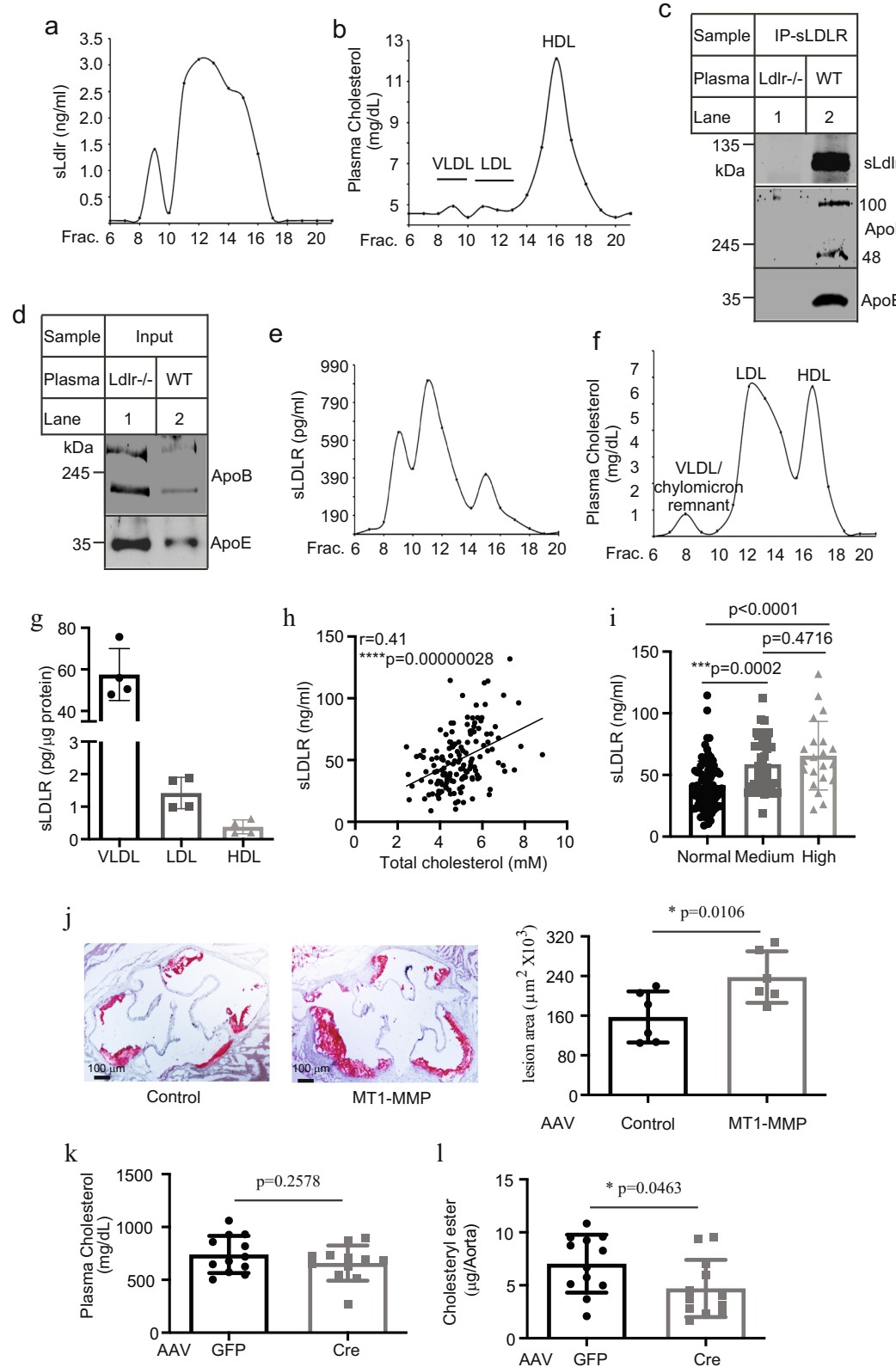

together, our findings show that MT1-MMP regulates LDLR levels via promoting ectodomain cleavage of the receptor. We noticed that knockout of hepatic MT1-MMP in mice markedly reduced the levels of plasma sLDLR by ~67% and significantly increased liver LDLR levels (Fig. 4d, e). This indicates that, at least in the liver, MT1-MMP is the proteinase mainly responsible for the ectodomain shedding of LDLR. Given the ubiquitous expression of LDLR, LDLR shedding in other tissues may contribute to plasma sLDLR detected in $MT1^{LKO}$ mice. However, we cannot rule out the possibility that other hepatic proteinases might also mediate LDLR shedding in the liver. For example, it has been reported that ADAM17 can promote LDLR shedding in HepG2 cells to a small extent[15]. More experiments are needed to define these possibilities.

**Fig. 6 Analysis of plasma sLDLR.** Profiles of plasma sLDLR (**a**) and lipoprotein cholesterol (**b**). Fasting mouse plasma was pooled from three mice and then applied to FPLC. Each sample was assayed in triplicate. Immunoprecipation (**c**) and immunoblotting (**d**). Plasma pooled from three $Ldlr^{-/-}$ or the wild-type mice was applied to a rat anti-mouse LDLR antibody and protein-G beads. The immunoprecipitated proteins (**c**) and 3 μl of pooled plasma (**d**) were subjected to immunoblotting. Representative images were shown. Similar results were obtained from three independent experiments. Profiles of human plasma sLDLR (**e**) and lipoprotein cholesterol (**f**). The experiment was performed as described in panel 6a and b except that fasting human plasma was used. Each sample was assayed in triplicate. **g** sLDLR in lipoproteins. sLDLR in purified human VLDL, LDL and HDL was measured and normalized to protein concentrations. Each sample was assayed four times. **h, i** Plasma levels of sLDLR and total cholesterol. sLDLR and total cholesterol in fasting human plasma samples were measured using kits. The association (**h**) was analyzed using with the Pearson's correlation coefficient and statistical significance among different groups (**i**) was analyzed with one-way Anova and Tukey post-hoc test using GraphPad Prism 9 (**i** $n = 87$ individuals in the normal group, 40 in the Medium group, and 21 in the high group). **j** Analysis of atherosclerosis. 8–10 week-old male apoE$^{-/-}$ mice were injected with AAV-Empty (Control) or AAV-MT1-MMP (MT1-MMP) and then fed the Western diet (6 mice per group). Atherosclerotic lesions were quantified using OMAX ToupView. Representative images were shown (Magnification 40×). Similar results were obtained from other mice. **k, l** MT1-MMP knockdown. 8–10-week-old male $MT1^{Flox}$/ApoE$^{-/-}$ mice were injected with AAV-GFP (GFP) or AAV-TBG-Cre (Cre) and then fed the Western diet (13 and 11 mice in the GFP and Cre group, respectively). Plasma cholesterol (**k**) and aortic cholesteryl ester (**l**) were measured. Student's $t$ test (two-sided) was carried out to determine the significant differences between groups (**j–l**). The significance was defined as $p < 0.05$. Values of all data were mean ± SD. Source data are provided as a Source Data file.

Hepatic LDLR is critical for the clearance of circulating apoB-100 and apoE-containing lipoproteins. Knockdown of MT1-MMP in cultured cells indeed enhanced cellular LDL uptake. Furthermore, $MT1^{LKO}$ mice, especially on the Western-type diet exhibited an increase in liver LDLR levels and a significant reduction in plasma levels of total cholesterol, HDL and non-HDL cholesterol. This reduction is likely caused by enhanced LDLR-mediated clearance of LDL, chylomicron remnants, as well as apoE-containing HDL particles as reported in PCSK9 knockout mice[5,33,39]. Consistently, overexpression of MT1-MMP in apoE$^{-/-}$ mice significantly increased plasma cholesterol levels. However, knockdown of MT1-MMP did not significantly affect plasma cholesterol level in apoE$^{-/-}$ mice. It is of note that majority of plasma cholesterol in apoE$^{-/-}$ mice is remnant cholesterol. apoE$^{-/-}$ mice do not express apoE, the ligand of LDLR. Thus, chylomicron and VLDL remnants cannot be cleared by LDLR. This may explain why knockdown of MT1-MMP significantly reduced plasma cholesterol levels in mice with the wild-type background but not in apoE$^{-/-}$ mice. Similarly, knockout of PCSK9 does not significantly affect plasma cholesterol levels in apoE$^{-/-}$ mice. It has been reported that overexpression of PCSK9 enhances the development of atherosclerosis and knockout of PCSK9 reduces the levels of aortic cholesterol in apoE$^{-/-}$ mice [28]. Consistently, we found that the atherosclerotic lesion area in the aortic sinuses was significantly increased in apoE$^{-/-}$ mice overexpressing MT1-MMP, while MT1-MMP knockdown reduced cholesteryl ester accumulated in the aorta of apoE$^{-/-}$ mice. Thus, MT1-MMP stimulates the development of atherosclerosis most likely through promoting hepatic LDLR shedding.

The shed ectodomains of VLDLR and apoER contain the ligand-binding domain and can bind to their ligand, Reelin, functioning as a competitive negative regulator to block binding of Reelin to the cell surface receptors[29,30,40]. Here, we also found that mouse plasma sLDLR co-eluted with VLDL, LDL and part of HDL on the size-exclusion chromatograph and co-immunoprecipitated with apoB and apoE. Similarly, the majority of human sLDLR were present in the fractions of VLDL and LDL. A small amount of human sLDLR was also present in the HDL fractions of human plasma, but to a much lesser extent as compared to mouse sLDLR. This might be caused by the fact that mouse HDL is more enriched in apoE compared to human HDL[41]. For example, knockout of PCSK9 in mice dramatically reduces plasma levels of LDL and HDL due to enhanced LDLR-mediated clearance of LDL and apoE-HDL[5,39]. On the other hand, inhibition of PCSK9 in human markedly reduces plasma levels of LDL but not HDL cholesterol[42]. It will be of interest to examine if sLDLR-bound LDL can be taken up by cell-surface LDLR as efficiently as sLDLR-free LDL.

MT1-MMP plays an essential role in tissue remodelling by cleaving extracellular matrix components and many non-extracellular matrix substrates[16,20,21,43]. However, compared to its roles in collagenolysis[44], our understanding of the non-extracellular matrix targets of MT1-MMP and their related physiological roles is poor. Recently, MT1-MMP has been shown to shed LRP1 in breast carcinoma MCF7 cells as well as in vascular smooth muscle cells and in cartilage[21,45,46]. However, we found that MT1-MMP had no detectable effect on LRP1 expression in cultured hepatocytes and mouse liver. It has been reported that MT1-MMP efficiently mediated LRP1 shedding in fibroblastoid type but not the epithelioid variant of HT1080 cells[21]. Thus, MT1-MMP appears to cleave LRP-1 in a cell-type-dependent manner.

Taken together, our findings uncover the fact that hepatic LDLR is cleaved by MT1-MMP. It is well documented that MT1-MMP is highly expressed in various types of cancer cells and promotes cancer metastasis and angiogenesis[16,44,47–50]. In addition, MT1-MMP is expressed in human atherosclerotic plaques and promotes plaque rupture by enhancing degradation of collagens[51]. We further demonstrated that overexpression of MT1-MMP increased the development of atherosclerosis while knockdown of MT1-MMP reduced aortic cholesteryl ester accumulation in apoE$^{-/-}$ mice. Collectively, these findings indicate that MT1-MMP may act as a shared risk factor for both cardiovascular disease and cancers, the two leading causes of global morbidity and mortality. Thus, inhibition of MT1-MMP is a very promising and valuable therapeutic target as it has the potential to increase hepatic LDLR levels, lower circulating LDL-C levels, increase atherosclerotic plaque stability, and reduce the risk of cancer metastasis and invasion.

## Methods

**Materials.** Minimum Essential Medium α (MEM α) without nucleosides, RMPI 1640 medium, Opti-MEM™, penicillin-streptomycin, trypsin-EDTA solution, Dil-labeled human LDL, unlabeled human LDL, Lipofectamine® 3000, Lipofectamine® RNAiMAX, High Capacity Complementary DNA (cDNA) Reverse Transcription Kit, SYBR®Select Master Mix, GeneJet and PureLink™ Hipure plasmid miniprep, Bicinchoninic acid (BCA) protein assay kit, and TRIzol® were obtained from ThermoFisher Scientific. HDL and LDL/very low-density lipoprotein (VLDL) Cholesterol Assay Kit were purchased from Cell Biolabs Inc. Total Cholesterol Assay Kit was from Wako Life Sciences. Human and mouse LDLR or PCSK9 DuoSet® or human LDLR Quantikine® ELISA kits were from R&D Systems. Alanine transaminase colorimetric activity assay kit was obtained from Cayman Chemical. RNeasy® Mini kit was from Qiagen. Dulbecco's Modified Eagle's Medium (DMEM), fetal bovine serum (FBS), bovine serine albumin (BSA), Complete™ EDTA-free protease inhibitors, and X-tremeGENE™ HP DNA

transfection Reagent were purchased from Millipore Sigma. All other reagents were obtained from Fisher Scientific unless otherwise indicated.

Recombinant full-length human PCSK9 with a FLAG tag at the C-terminus was purified from culture medium of human embryonic kidney (HEK)-293S cells stably expressing PCSK9 using ANTI-FLAG® M2 affinity gel (Sigma) and FLAG peptide (Sigma) as described in our previous studies[6–8,52]. The following antibodies were used: a rabbit anti-LDLR polyclonal antibody, 772B; HL-1, a mouse monoclonal anti-the linker sequence between ligand binding repeat (LR) 4 and LR5 of LDLR antibody[7,53]; 3143, a rabbit anti-LDLR polyclonal anti-serum directed against the C-terminal 14 amino acid residues of LDLR[7,54]; 15A6, a mouse anti-PCSK9 monoclonal antibody[52]; mouse anti-MT1-MMP monoclonal antibodies (Millipore, clone LEM-2/15.8, MAB3329); a rabbit anti-MT1-MMP monoclonal antibody (Abcam, ab51074); a rabbit anti-MT2-MMP polyclonal antibody (ThermoFisher, PA5-13184); a rabbit anti-LRP1 polyclonal antibody (Novus Biologicals, NBP1-40726); a mouse anti-calnexin monoclonal antibody (BD Biosciences, 610524); a mouse anti-actin monoclonal antibody (BD Biosciences, 612657); a mouse anti-transferrin receptor monoclonal antibody (BD Biosciences, 612125); Dylight™ 800 and Dylight™ 680 conjugated rabbit anti-hemagglutinin epitope (HA) antibody (Rockland Immunochemicals, 600-445-384 and 600-444-384); a mouse anti-Myc monoclonal antibody 9E10 that was purified from culture medium from hybridoma cells obtained from the American Type Culture Collection (CRL-1729) using Protein G Sepharose Fast Flow (GE Healthcare).

**Animal.** C57BL/6J and apoE$^{-/-}$ mice were purchased from The Jackson Laboratory, housed and bred in the animal facility at the University of Alberta. The Cre-lox strategy was used to selectively inactivate MT1-MMP in mouse liver. The vector, in which exon 2 and exon 4 of mouse MT1-MMP gene were flanked by LoxP sites, was purchased from EUCOMM. MT1-MMP$^{FRT-Flox}$ mice were generated in the Clara Christie Centre for Mouse Genomics at the University of Calgary using embryonic stem cells that have 50% C57 background. The homozygous MT1-MMP$^{FRT-Flox}$ was crossed to the FLPo mice (The Jackson Laboratory) to remove the FRT-flanked selection marker (Fig. S3a). The resulting MT1$^{Flox}$ mice were then backcrossed with C57BL/6J for 6 times and then crossed with transgenic mice expressing Cre recombinase under the control of the albumin promoter (The Jackson Laboratory) to produce mice with no active MT1-MMP in hepatocytes (MT1$^{LKO}$). Genotyping was performed by PCR using AccuStart™ II mouse genotyping kit (Quanta Biosciences, Beverly, MA) and three primers that were designed with Primer3 and synthesized by Integrated DNA Technologies (IDT®, Coralville, CA) (Supplementary Table 1).

Mice were housed 3–5 per cage with free access to H$_2$O in a climate-controlled facility (22 °C, 43% humidity) with a 12 h light/dark cycle. After weaning, mice were fed chow diet ad libitum containing 20% protein, 5% fat, and 48.7% carbohydrates (LabDiet, PICO Laboratory Rodent Diet 20, gross energy 4.11 kcal per gm). For the Western diet experiment, mice were fed the Western Diet containing 0.15% cholesterol from TestDiet or Research Diet Inc (kcal from fat 40%, protein 16%, and carbohydrate 44%). All animal procedures were approved by the University of Alberta's Animal Care and Use Committee (protocol number AUP00000456) and were conducted in accordance with guidelines of the Canadian Council on Animal Care.

**Site-directed mutagenesis, cell culture and transfection.** Plasmid pCR3.1 containing cDNA of the full-length MT1-MMP with an HA-tag between Asp115 and Glu116 (pCR3.1-MT1-MMP, a kind gift from Dr. Weiss, University of Michigan) was used to generate the mutant form of MT1-MMP. Mutagenesis was performed using the QuickChange Lightning site-directed mutagenesis kit (Agilent Technologies) according to the manufacturer's instruction[8,55–58]. The sequences of the oligonucleotides containing the residues to be mutated were synthesized by IDT® and listed in Supplementary Table 1. The presence of the desired mutation and the integrity of each construct were verified by DNA sequencing.

All cells were cultured at 37 °C in a 5% CO$_2$ humidified incubator. DsiRNA and plasmid DNA were introduced into cells using Lipofectamine™ RNAiMAX and X-tremeGENE™ HP (HEK293, Huh7.5, and Hepa1c1c7 cells) or Lipofectamine™ 3000 (HepG2 cells), respectively, according to the manufacturer's instruction[59]. Scrambled and predesigned DsiRNAs against MT1-MMP were purchased from IDT® and listed in Supplement Table 1. 2 μl of 20 μM siRNA solution was added to 100 μl of Opti-MEM medium and vortexed (Solution A) (one well of a 6-well plate). 6 μl of RNAiMAX reagent was then added to 100 μl of Opti-MEM medium and vortexed (Solution B). Solution A and B were then combined, vortexed and incubated for 5 min at room temperature. The solution mixture was then added to cells drop by drop. For X-tremeGENE HP transfection, 2 μg of DNA was added to 200 μl of serum reduced Opti-MEM and vortexed briefly (one well of a 6-well plate). Next, 2 μl of X-tremeGENE HP DNA transfection reagent was added and vortexed. The transfection mixture was then incubated for 30 min at room temperature before adding drop by drop to cells. For Lipofectamine 3000 transfection, 2 μg of DNA and 5 μl of P3000 reagent was added to 125 μl serum reduced Opti-MEM medium and vortexed (Transfection mixture A) (one well of a 6-well plate). 5 μl of Lipofectamine 3000 reagent was then added to 125 μl of Opti-MEM medium and vortexed (Transfection mixture B). Transfection mixture A and B were then combined and vortexed briefly, followed by a 5 min incubation at room temperature. The Mixture was then added drop by drop to cells. 48 later, the cells and medium were collected for analysis.

HEK293, HepG2, Huh7.5 and McA-RH7777 cells were maintained in DMEM (high glucose) containing 10% (v/v) FBS. Hepa1c1c7 cells were maintained in MEM α (no nucleotides) containing 10% FBS. Primary human hepatocytes (Triangle Research Labs) were thawed from liquid nitrogen stocks and seeded onto a 12-well collagen-coated plate at the density of 2 × 10$^5$ cells per well in 1 ml of RPMI containing 10% FBS[60]. Cells were incubated at 37 °C and allowed to attach to the plate for 4 h before siRNA transfection or adenovirus infection. 48 h after treatment, the cells were lysed in lysis buffer A (1% Triton, 150 mM NaCl, 50 mM HEPES, pH 7.4) containing 1 × Complete EDTA-free protease inhibitors for 30 min on ice. After centrifugation for 15 min at 20,000 × g at 4 °C, the supernatant was collected and protein concentrations were determined by the BCA protein assay. Same amount of whole-cell lysate was subjected to immunoblotting. Mice primary hepatocytes were isolated as described[61]. Briefly, the liver was perfused with HBSS buffer containing 0.5 mM EGTA and then HBSS containing 1mg/ml of collagenase for 6–10 min. The isolated hepatocytes were seeded on collagen-coated dishes in DMEM (high glucose) containing 0.5% FBS for up to 48 h.

**Inhibitor treatment.** When Huh7 cells reached to 90% confluence in 6 well plates, culture medium was removed and replaced with medium containing GM6001 (DMEM + 10% FBS medium at 30 and 100 μM final concentration of GM6001). Cells were then incubated for 16 h prior to lysis for western blot analysis.

Huh7 cells were transfected with empty pCDNA3.1 or HA-tagged MT1-MMP-pCDNA3.1. 48 h later, the cells were incubated with MG132 (10 μM, MG) or chloroquine (10 μM, Chloro) for 6 h. After, cells were washed once in PBS and then harvested for the preparation of whole-cell lysate for western blot analysis.

**Immunoblot and immunoprecipitation analysis.** Whole-cell lysate was prepared from cultured cells 48 h after transfection[59]. Culture medium was removed from plates and cells were washed with 1 ml of PBS. The cells were scraped, collected in 1 ml of PBS and then centrifuged at 4 ºC for 5 min (1500 × g). The supernatant was discarded. Cells were lysed in lysis buffer A (1% Triton, 150 mM NaCl, 50 mM HEPES, pH 7.4) containing 1 × Complete EDTA-free protease inhibitors for 30 min on ice and vortexed intermittently every 10 min. Cell lysis was spun for 10 min at 20,000 × g at 4 °C, the supernatant was collected as whole-cell lysate. Protein concentrations were determined by the BCA protein assay.

Snap frozen liver tissue samples stored at −80 °C were thawed and homogenized using PowerGen 500 Homogenizer (Fischer Scientific) in a buffer (250 mM Sucrose, 50 mM Tris-HCl, pH7.4, 1 mM EDTA, 1%Triton X-100, and protease inhibitors). Homogenized tissue was then incubated on ice for 30 min with intermittent vortex in every 10 min. After centrifugation for 15 min at 20,000 × g at 4 °C, the supernatant was collected and protein concentrations were determined by the BCA protein assay. Same amount of total proteins from tissue homogenate or whole-cell lysate was subjected to SDS polyacrylamide gel electrophoresis (SDS-PAGE, 8%) and transferred to nitrocellulose membranes (10600004, GE Healthcare) by electroblotting. Immunoblotting was performed using specific antibodies as indicated. Antibody binding was detected using IRDye®680 or IRDye®800-labeled goat anti-mouse or anti-rabbit IgG (Li-Cor). The signals were detected on a Licor Odyssey Infrared Imaging System (Li-Cor). All image figures were adjusted in Adobe Photoshop 2020 and them made in Adobe Illustrator 2020. The color of most images was discarded in Photoshop. Contrast and exposure were adjusted equally cross the whole image in Photoshop for better visualization. The sizes of images were adjusted in Illustrator.

Immunoprecipitation was performed using whole-cell lysate[57]. Cells were cultured in DMEM containing 5% newborn calf lipoprotein-poor serum (NCLPPS) for 16 h to increase expression of LDLR and then lysed in lysis buffer A containing 1× Complete EDTA-free protease inhibitors. Same amount of total proteins was applied to a monoclonal anti-LDLR antibody (HL-1), a monoclonal anti-MT1-MMP antibody (Millipore, clone LEM-2/15.8, MAB3328), or a monoclonal anti-Myc antibody (9E10) and protein G beads to immunoprecipitate LDLR or MT1-MMP. Immunoprecipitated samples were washed three times with lysis buffer A. The immunoprecipitated proteins were then eluted from the beads by addition of 2 × SDS-PAGE sample buffer (100 mM Tris-HCl, pH 6.8, 4% SDS, 20% glycerol, 0.04% bromophenol) containing 5% β-mercaptoethanol. The eluted samples and whole-cell lysate were subjected to SDS-PAGE (8–20%) and immunoblotting.

**Quantitative real-time PCR.** Total RNAs were extracted from cultured cells and mouse tissues using the RNeasy® Mini kit and TRIzol®, respectively, according to the manufacturer's instruction[59]. cDNA was synthesized using the High Capacity cDNA Reverse Transcription Kit. Relative qRT-PCR was carried out on StepOnePlus™ using SYBR®Select Master Mix according to the manufacturer's instruction. Each sample was processed in triplicate, and the average cycle threshold was calculated. Relative gene expression was normalized to the glyceraldehyde-3-phosphate dehydrogenase gene (GAPDH) that had a similar amplification efficiency as that of the target genes. Primers for human and mouse GAPDH, MT1-, MT2-, MT3-, MT4-, MT5-, MT6-MMP, HMGCR, LDLR, PCSK9,

*SREBP2* and *IDOL* were designed by PrimerQuest Real-Time PCR Design Tool, synthesized by IDT, Inc. and listed in Supplement Table 1.

**PCSK9-promoted LDLR degradation.** Huh7.5 cells seeded in a 12-well plate ($1.5 \times 10^5$ cells per well) were transfected with scrambled or MT1-MMP siRNA. 48 h after, cells were washed twice with DMEM, followed by a 4-h incubation with PCSK9 (2 µg per ml) in 0.5 ml of DMEM containing 5% NCLPPS. The cells were then collected for the preparation of whole-cell lysate. Protein concentrations were determined by the BCA protein assay. Same amount of whole-cell lysate was subjected to immunoblotting.

**Binding of Dil-LDL to LDLR.** Huh7.5 cells were seed at a density of $3 \times 10^4$ cells per well in a 96-well plate in 100 µl of DMEM containing 10% FBS. 24 h later, the cells were transfected with scrambled or MT1-MMP siRNA. 48 h after transfection, cells were washed with Opti-MEM. Dil-labeled LDL (10 µg per ml) was then added to cells in 100 µl of DMEM containing 5% NCLPPS in the presence or absence of unlabeled human LDL (600 µg per ml)[62,63]. The plates were incubated at 37 °C for 6 h. After, the cells were washed four times in a washing buffer (50 mM Tris-HCl, pH7.4, 150 mM NaCl, 2 mg per ml BSA) and then lysed in 100 µl of RIPA buffer (50 mM Tris-HCl, pH7.4, 150 mM NaCl, 1% Triton X-100, 0.5% sodium deoxycholate, 0.1% SDS) contacting 1× protease inhibitors. The lysate was then transferred to a 96-well black plate for the measurement of fluorescence using a SYNERGY plate reader (Excitation: 520 nm; Emission: 580 nm). The concentrations of total proteins in each well were measured using the BCA protein assay. LDL uptake was calculated by normalization of the fluorescence units to the amount of total proteins in the same well. The results obtained in the presence of excess unlabeled LDL revealed the nonspecific binding. Specific binding was calculated by subtraction of nonspecific binding from the total counts measured in the absence of unlabeled LDL.

**Biotinylation of cell surface proteins.** Huh7.5 cells were seeded in a 6-well plate in 2 ml of culture medium at the density of $2.5 \times 10^5$ cells per ml. 24 h later, cells were transfected with scrambled or MT1-MMP siRNA. 24 h after transfection, cells were cultured in DMEM containing 5% NCLPPS for 16 h. Cell surface proteins were then biotinylated with EZ-Link Sulfo-NHS-LC-Biotin (Pierce) in PBS (0.5 mg per ml, pH 8.0) for 15 min at 4 °C[64]. After quenching with 100 mM glycine, the cells were lysed in 150 µl of lysis buffer A. A total of 50 µl of cell lysate was saved and ~100 µl of the lysate (same amount of total proteins) were added to 60 µl of 50% slurry of Neutravidin agarose (Pierce) to pull down biotinylated cell surface proteins, which were then eluted from the beads by adding 1 × SDS loading buffer and analyzed by immunoblotting.

**Immunofluorescence.** Huh7.5 or HepG2 cells seeded onto coverslips ($1.0 \times 10^5$ cells per ml) were transfected with or without MT1-MMP cDNA as indicated[8,56,64]. 36 h later, cells were cultured in DMEM containing 5% NCLPPS for 16 h, fixed with 3% paraformaldehyde and then permeabilized with cold methanol for 10 min at -20 °C. The cells were then incubated with a mouse anti-LDLR monoclonal antibody and a rabbit anti-MT1-MMP (Abcam) (1:100) or anti-HA antibody. Antibody binding was detected using Alexa Fluor 488 goat anti-rabbit or mouse IgG and Alexa Fluor 568 goat anti-mouse or rabbit IgG. Nuclei were stained with 4', 6-diamidino-2-phenylindole (DAPI, ThermoFisher). After washing, coverslips were mounted on the slides with ProLong Diamond Antifade Mountant (ThermoFisher). Localizations of LDLR and MT1-MMP were determined using a Leica SP5 laser scanning confocal microscope (filters: 461 nm for DAPI, 519 nm for Fluor 488, and 543 nm for Fluor 568).

**Immunohistochemistry.** All liver sectioning and staining were performed by the HistoCore facility in Alberta Diabetes Institute at the University of Alberta. Briefly, a portion of livers was fixed in 10% formalin and embedded in paraffin, followed by a section at 5 µm thickness onto Histobond slides. The slides were then subjected to hematoxylin and eosin (H&E) or Trichrome staining. Cryosectioned liver tissues were used for Oil Red O staining. The resulting slides were imaged using ZEISS Axio Observer A1 in triplicate. Relative stained area was then quantified with ImageJ software (1.52S, National institute of Health) using color segmentation and threshold analysis.

**Extraction of lipids.** Lipids were extracted from the liver and the aorta using the Folch method[65]. Liver samples were homogenized in 50 mM Tris-HCl (pH7.4), 250 mM sucrose, and 1 mM EDTA by a glass/Teflon homogenizer for 20 s. Lipids were extracted from 4 mg of liver homogenate with chloroform:methanol (2:1). The aorta was cut into small pieces and homogenized in 250 µl of 50 mM Tris-HCl (pH7.4), 250 mM sucrose, and 1 mM EDTA by a glass/Teflon homogenizer. Samples were sonicated 3 × 10 s on ice and the volume was completed to 1 ml with PBS. Homogenized samples were transferred into a glass tube with 4 ml of chloroform/ methanol (2:1) and incubated for 1 h with vigorous agitation. The extraction from liver or aorta homogenate was spun at $1000 \times g$ for 10 min, the bottom phase was transferred to a new glass tube and dried under nitrogen. Lipids were dissolved in 1 ml or 500 µl of chloroform with 2% Triton X-100 for the liver

and the aorta homogenate, respectively. The samples were dried under nitrogen and redissolved in 1 ml or 250 µl of distilled H₂O for the liver and the aorta samples, respectively. TG and total cholesterol in 10 µl of the liver samples were measured using commercial kits from Roche Diagnostics and Wako Diagnostics, respectively. For the aorta samples, 30 µl of lipid extract was loaded per well on a 96-well plate together with a set of known cholesterol standards for the measurement of total cholesterol and free cholesterol using commercial kits from Cell Biolabs. Cholesteryl ester was determined by deducting the values of free cholesterol from that of total cholesterol.

**Plasma lipid, ALT, and sLDLR analysis.** Blood samples were collected into heparin-coated tubes (BD Biosciences) from tail or saphenous veins of mice (age of 8–12 weeks) on a chow or the Western-Type diet. The blood was centrifuged at $3000 \times g$ for 20 min for plasma isolation. 10 or 20 µl of plasma from each mouse were subjected to ALT or total cholesterol measurement using the ALT colorimetric activity assay kit and the Cholesterol E kit (Wako Diagnostics), respectively, according to the manufacturer's instructions. Plasma HDL and non-HDL were separated using the HDL and LDL/VLDL Cholesterol Assay kit. 5 µl of plasma were mixed with 5 µl of the Precipitation Reagent, centrifuged at $2000 \times g$ for 20 min. The supernatant (HDL fraction) was transferred to a new tube. The pellet was dissolved in 10 µl of PBS. Cholesterol content in each fraction was measured using the Cholesterol E kit. Lipoprotein profiles were analyzed by the Lipidomic Core Facility at the University of Alberta. Briefly, 5 µl of plasma from each mouse in the same experiment group was pooled and subjected to Fast Protein Liquid Chromatography (FPLC) with a Superose 6 column.

sLDLR in culture medium was measured by both western blot and ELISA. Cells were transfected with siRNA or plasmid vectors as described above. 24 h later, cells were switched to serum free medium for 16 h. After collection, culture medium was subjected to trichloroacetic acid (TCA) precipitation (15% final TCA concentration) overnight at 4 °C. After a centrifugation at $20,000 \times g$ for 20 min at 4 °C, the pellets were washed once in cold acetone (−20 °C), dried and resolubilized in 9 M urea. Protein concentrations were measured using the BCA assay. Equal amount of total proteins was analyzed by immunoblotting. sLDLR levels in culture medium (1:5 diluted in the Reagent Diluent) and plasma from each mouse (1:20 diluted in the Reagent Diluent) were assessed using the Human or Mouse LDLR DuoSet ELISA assay kit in accordance to manufacturer's protocol (R&D Systems). Briefly, standards and diluted samples were added to each well in a 96-well plate that was coated with the Capture Antibody and blocked in Reagent Diluent. After incubating for 2 h at room temperature, the plate was washed and then incubated with the Detection Antibody, followed by Streptavidin-HRP, Substrate Solution and Stop Solution. The optical density of each well was measured using a SYNERGY plate reader at 540 nm. Plasma sample from $Ldlr^{-/-}$ mice was used as a negative control for mouse plasma samples. Specific levels were calculated by subtraction of counts of plasma samples of $Ldlr^{-/-}$ mice from that of $MT1^{LKO}$ and $MT1^{Flox}$ mice.

**Human plasma sLDLR and lipoprotein cholesterol.** 148 subjects were recruited for the collection of overnight fasting blood samples at the Taishan Medical University. Participants were randomly recruited during their normal physical exam. All patient information was obtained based on the questionnaire. Inclusion criteria was healthy adults (20+ years) without pre-existing health problems. No exclusion was applied. All participants did not report health problems. EDTA-plasma was isolated, aliquoted and stored at −80 °C. Plasma triglycerides (TG), LDL-C and total cholesterol were measured in the clinical laboratory of the Taishan Medical University. Plasma levels of sLDLR and PCSK9 were determined using the human LDLR and PCSK9 DuoSet ELISA assay kit according to manufacturer's instruction (R&D system), respectively. All participants provided written informed consent and the procedure was approved by the Research Ethic Committee in Institute of Atherosclerosis at Taishan Medical University (Shandong First Medical University).

Plasma lipoprotein profile was assessed by gel filtration. Briefly, 500 µl of pooled normal fasting human heparin-plasma (Innovative Research Inc. Novi, MI) or pooled fasting mouse plasma (C57BL/6J) was applied to a size exclusion (Superose 6 10/300 GL)-FPLC column on an AKTA purifier system (GE Healthcare Life Sciences) and then eluted with 1 x PBS. The eluate was collected in 1 mL fractions. The levels of total cholesterol and sLDLR in each fraction were measured using the Cholesterol E kit (Wako Diagnostics) and the Human or Mouse LDLR DuoSet ELISA assay kit (R&D system), respectively.

**Gelatin zymography.** Culture medium was collected from cells 48 h after treatment and protein concentrations were measured by the BCA assay. Same amount of culture medium (80 µg of total proteins) or 5 µl of plasma sample from each mouse was used for zymography analysis as described[23]. Briefly, samples were mixed with 4 × denaturing sample buffer (200 mM Tris-HCl, pH 6.8, 8% SDS, 40% glycerol, 0.08% bromophenol) and subjected to electrophoresis on an 8% SDS gel containing 2 mg/ml of gelatin. After, the gel was washed 3 times with 2.5% Triton X-100 in H₂O and then incubated for 16–20 h in a renaturing buffer (50 mM Tris-HCl, pH7.5, 5 mM CaCl₂, 150 mM NaCl, and 0.05% NaN₃). Gels were stained with Coomassie Blue R250, destained in 40% (v/v) methanol with 10% (v/v) acetic acid

and then scanned on a Licor Odyssey Imaging System. MMP2 was visualized as a clear band of degraded gelatin on the blue background of Coomassie Blue staining.

**Adenovirus and adeno-associated virus**. Recombinant adenoviral vectors containing cDNAs for the wild-type and mutant human MT1-MMP were generated using the AdEasy™ Adenoviral Vector System (Agilent Technologies)[66]. MT1-MMP cDNA was subcloned into the pShuttle-CMV vector between the restriction enzyme sites HindIII and EcoRV, followed by in vitro cre-lox recombination. Adenoviruses were packaged and amplified in QBI 293 cells and then purified using CsCl density-gradient ultracentrifugation. After desalting with the EconoPac 10DG column (Bio-Rad), adenoviral particles were injected into mice intravenously ($1.0 \times 10^{11}$ particles per mouse). 72 h later, tissues and blood were collected from euthanized mice. For cultured cells, cells were plated on a 12-well plate at the density of $1.5 \times 10^5$ cells per well in 1 ml of culture medium. 24 h later, cells were infected with adenovirus ($4 \times 10^9$ particles per well). 48 h after infection, the cells were collected for analysis.

AAV carrying cDNA for human MT1-MMP or cre recombinase under the control of a hepatocyte-specific thyroxine-binding globulin (TBG) promoter was generated using the AAV-DJ/8 Helper Free Expression Complete System in accordance with the manufacturer's instruction (Cell Biolabs, Inc). pAAVTBG.PI. EGFP.WPRE.bGH (Addgene) was cut with NotI and HindIII and a fragment of about 4.8 kb was isolated. cDNA for human MT1-MMP and cre recombinase was amplified from plasmid pCR3.1-MT1-MMP and adenovirus expressing cre with PCR using a forward primer with the Not I site in the begining and a reverse primer containing the Hind III site at the end, respectively. The PCR fragment was digested with NotI and Hind III, purified and then ligated to NotI-HindIII digested pAAVTBG.PI.EGFP.WPRE.bGH using the Quick ligation kit (NEB). The integrity of the cDNA of MT1-MMP and cre was verified by DNA sequencing. The resulting vector together with pAAV-helper and pAAV-DJ/8 was transfected to QBI 293A cells using polyethylenimine (PEI. Polysciences, Warrington, PA). 72 h post-transfection, AAV was purified from the cells using OptiPrep™ (Sigma) density-gradient ultracentrifugation as described[67]. AAV particles were collected from the 40% density step, diluted in PBS and concentrated with Amicon Ultra-15 Centrifugal Filter Unit (Millipore, 100K NMWL). AAV titers were determined using qRT-PCR. Purified AAV particles were injected into mice intravenously ($1.0 \times 10^{11}$ genomic copy (gc) per mouse). Mice were euthanized at the indicated time after injection. Blood and tissues were collected for analysis.

**Atherosclerotic analysis**. For the overexpression experiment, 8–10-week-old male apoE$^{-/-}$ mice were purchased from the Jackson Laboratory and housed in the animal facility at the University of Alberta. The mice were randomly divided into two groups (11-13 mice per group) and injected with AAV-Empty (the control group) or AAV-MT1-MMP ($1.0 \times 10^{11}$ gc per mouse). The mice were then fed the Western Diet from TestDiet (5TJN) that containing 0.15% cholesterol, kcal from fat 40% (milk fat, lard and soybean oil), protein 16%, and carbohydrate 44%) for 8 weeks. For the knockdown experiment, we developed apoE$^{-/-}$/MT1$^{Flox}$ mice via crossing $MT1^{Flox+/+}$ mice with apoE$^{-/-}$ (The Jackson Laboratory) at the University of Alberta. Male apoE$^{-/-}$/MT1$^{Flox}$ mice (8–10 weeks old) were randomly divided into two groups (11 mice per group) and injected with AAV-GFP (the control group) or AAV-TBG-Cre ($1.0 \times 10^{11}$ gc per mouse). The mice were then fed the Western Diet from the Research Diet Inc (Catalogue # D12079Bi) that containing 0.15% cholesterol, kcal from fat 40% (anhydrous butter and corn oil), protein 16%, and carbohydrate 44% for 8 weeks. At the end point, the aortas and hearts were collected immediately from euthanized mice and fixed in 4% paraformaldehyde. Serial sections (8 μm thick) were taken throughout the three aortic valves of each mouse and six sections per mouse were collected for the analysis. Images were taken using an OMAX M837ZL-C140U3 microscope (Magnification 100×). The atherosclerotic burden was quantified by measuring the surface area of Oil Red O positive lesions on the cross-sectional area of the aorta sinus. Lesion areas were quantified with OMAX ToupView.

**Statistical analysis**. All statistical analyses were carried out by Prism version 9.0 (GraphPad Software). Student's t test or one-way ANOVA with Tukey post-hoc test was carried out to determine the significant differences between groups. All data met normal distribution criteria and variance between groups that was analysed by F-test showed no significant difference ($p > 0.05$). Correlation was analyzed using Pearsom correlation coeeficients (One-tailed). Values of all data unless otherwise indicated were mean ± S.D. The significance was defined as *$p < 0.05$, **$p < 0.01$, ***$p < 0.001$, ****$p < 0.0001$. All experiments except for where indicated were repeated at least three times.

**Reporting summary**. Further information on research design is available in the Nature Research Reporting Summary linked to this article.

## Data availability
Source data for all figures are provided with this paper and the Supplementary online data. Original data and uncropped versions of blots are provided as an online supplementary Source Data file. Source data are provided with this paper.

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

## Acknowledgements

This work was funded mainly by a Grant-in-Aid from Heart and Stroke Foundation of Canada (G-16-00012559) and by a project grant from Canadian Institutes of Health Research (CIHR PS 155994). X.X. and G.W. were supported by funding from Qingyuan People's Hospital. S.Q. was supported by 91539114, NSFC 81929002, as well as ts201511057. The authors are grateful to Drs. Helen Hobbs and Jonathan Cohen (University of Texas Southwestern Medical Center at Dallas), Richard Lehner and Carlos Fernandez-Patron (University of Alberta) for helpful discussions. We also thanked Dr. Stephen J. Weiss (Division of Molecular Medicine and Genetics, University of Michigan, MI, USA) and Dr. Zhongjun Zhou (The School of Biomedical Sciences, the University of Hong Kong, Hong Kong, China) for the gift of valuable biological samples and Dr. Weiss for the plasmid containing MT1-MMP cDNA. The authors thanked Drs. Liang Li, Jelske van der Veen, Paul LaPointe, and Sereana Wan for their technique supports as well as Audric Moses and the Lipidomics Core Facility of the Faculty of Medicine and Dentistry and the Women and Children's Health Research Institute at the University of Alberta for plasma lipid profile analysis, Greg Plummer and the Cellular Imaging Centre of the Faculty of Medicine and Dentistry at the University of Alberta for confocal microscopy.

## Author contributions

D.W.Z. designed and performed the experiments, collected and analyzed data, and supervised and directed this project. A.A., X.X., H.M.G., F.W. and N.Y. performed the experiments, collected and analyzed data. N.Y., Y.X., L.C. and S.Q. collected, processed and analyzed human plasma samples. D.N.D., N.M.K. and G.W. provided technical support and guidance, helpful discussions and comments. A.A. and D.W.Z. wrote the manuscript.

## Competing interests

The authors declare no competing interests.
