## [Peer Review File · Nature Communications]

Reviewers' Comments:

Reviewer #1:

Remarks to the Author:

This article describes in great detail how MT1-MMP can cleave the LDL receptor in liver cells. The role of this proteinase is suggested to play an important role in lipid metabolism – in general, but it has been known for long that LDLR's ectodomain can be clipped of by many of proteolytic enzymes such as metalloproteinases like MT1-MMP. However, such detailed focus on MT1-MMP has not been reported. This paper, although describing new data on MT1-MMP and processing of LDLR in hepatocytes does not provide novel conceptual understanding of lipid metabolism. Various metalloproteinases have been shown to cleave ectodomains of cellular receptors and thus influence many novel biological roles of receptor. A large body of data is shown to demonstrate the effects of MT1-MMP. The authors should provide for the editors and reviewers pictures of raw data (gesl) as it is difficult to evaluate the results from small gel segments.

Reviewer #2:

Remarks to the Author:

Alabi and coworkers report that membrane type 1-matrix metalloproteinase (MT1-MMP) promotes ectodomain cleavage of the low-density lipoprotein receptor and thus define a new role of membrane type 1 matrix metalloproteinase in lipid metabolism. Membrane type 1-matrix metalloproteinase (MT1-MMP) is a Zn²⁺-dependent endopeptidase that can cleave extracellular matrix and non-matrix substrates, such as LDLR. The major findings of this study include: 1) knockdown of MT1-MMP increased cellular LDLR abundance and reduced the levels of cleaved soluble ectodomain of LDLR (sLDLR) in cultured hepatocytes; 2) LDLR may interact with MT1-MMP, which is demonstrated by using co-immunoprecipitation and confocal microscopy; 3) mice lacking hepatic MT1-MMP displayed an increase in liver LDLR levels and a reduction in plasma levels of sLDLR, HDL-cholesterol, and non-HDL cholesterol; 4) the majority of sLDLR were associated with apolipoprotein B (apoB) and apoE-containing lipoproteins in both mouse and human plasma. The authors demonstrate that MT1-MMP promotes the ectodomain shedding of LDLR and plays an important role in lipid metabolism. These data implicate that MT1-MMP promotes the ectodomain shedding of LDLR and is crucial for lipid metabolism. It is an interesting study, however, there are several issues to be addressed.

Specific Comments:

1. The findings that MT1-MMP post-translationally mediates LDLR cleavage and regulates lipid metabolism are very interesting and potentially important. To enhance the significance of this study, it would be important to demonstrate the role of liver specific knockdown or overexpression of MT1-MMP in the development of atherosclerosis using a hypercholesterolemic mouse model.
2. Fig 1a. Having a schematic of the original hypothesis above the data showing that the hypothesis was incorrect is something of a distraction. The schematic of the hypothesis could be eliminated as it is stated in the text. Alternatively, please separate the hypothesis from the results of Figure 1a by putting them in separate panels.
3. There is a clear dose dependent relationship between MT1-MMP transfection and reduced expression of LDLR in Fig 1d. It would be important to determine the expression of PCSK9 in these samples.
4. "Addition of PCSK9 reduced cellular LDLR levels in scrambled siRNA transfected cells, as well as in cells transfected with MT1-MMP siRNA (Fig. 1b). Thus, knockdown of MT1-MMP did not affect PCSK9-promoted LDLR degradation." It seems MT1-2 siRNA transfection indeed increased PCSK9

expression and increased LDLR without addition of PCSK9 in Figure 1b. But PCSK9 is a potent way to degrade the LDLR. It might be important to confirm the results with the co-transfection of PCSK9 and MT1-2 in this experiment rather than addition of exogenous PCSK9.

5. The background of the Cal blot is too dark in Figure 1b. Please adjust the background to the same as others in Figure 1b.

6. Figure 5 d,e and f. The sample number is very small for a mouse study (n=3). The data would be strengthened by increasing the n. It would be important to quantify the TG levels and cholesterol levels in the livers in Figure 5f.

7. Please include the scale bar in Fig 3e, Fig 5e and Fig 5f.

8. Mice lacking hepatic MT1-MMP displayed an increase in liver LDLR levels and a reduction in plasma levels of sLDLR, HDL-cholesterol, and non-HDL cholesterol. It would be important to determine the expression levels of PCSK9 in these mice.

9. Specific knockout of MT1-MMP in mouse hepatocytes increased hepatic LDLR levels and reduced plasma levels of lipoprotein cholesterol. Although it has been shown LDLR levels were markedly increased in MT1-MMP knockdown cells in the absence of PCSK9 in human hepatocytes cell lines, does knockout of MT1-MMP affect the expression levels of PCSK9 in mouse hepatocytes?

10. In Figure 2, "(a) Inhibitors treatment. Huh7 cells transfected with empty pCDNA3.1 or HA-tagged MT1-MMP-pCDNA3.1 were incubated with MG132 (10 μ M, MG) or chloroquine (10 μ M, Chloro) for 6 h." and "48 h later, whole cell lysate was prepared and applied to immunoblotting with antibodies as indicated. TFR, transferrin receptor." Please use complete sentences in this Figure Legend and the others as well.

11. Figure 6. It is very interesting that the cholesterol levels are related to the soluble LDLR concentrations in humans. It would be important to measure PCSK9 levels in these samples. Is it possible to measure PCSK9 levels in these samples, or to obtain other samples that have been handled appropriately so the PCSK9 levels can be measured in the same samples with LDL-C and sLDLR? Please address this point in the discussion as well.

Minor:

1. "sLDLR levels in culture medium (1:5 diluted in the Reagent Diluent) and plasma from each mouse (1:20 diluted in the Reagent Diluent) were assessed using the Human or Mouse LDLR DuoSet ELISA assay kit in accordance to manufacturer's protocol." Please include the kit information (company).

2. Please include the scale bar of figure 3e, 5e and f in figures and describe in the figure legend.

3. The manuscript would benefit from careful editorial work. There are a number of distracting grammatical errors. (Page 5. To confirmed this finding,.. To confirm this finding,..)

Reviewer #3:

None

Reviewer #1: This article describes in great detail how MT1-MMP can cleave the LDL receptor in liver cells. The role of this proteinase is suggested to play an important role in lipid metabolism – in general, but it has been known for long that LDLR's ectodomain can be clipped of by many of proteolytic enzymes such as metalloproteinases like MT1-MMP. However, such detailed focus on MT1-MMP has not been reported. This paper, although describing new data on MT1-MMP and processing of LDLR in hepatocytes does not provide novel conceptual understanding of lipid metabolism. Various metalloproteinases have been shown to cleave

ectodomains of cellular receptors and thus influence many novel biological roles of receptor. A large body of data is shown to demonstrate the effects of MT1-MMP. The authors should provide for the editors and reviewers pictures of raw data (gesl) as it is difficult to evaluate the results from small gel segments.

Response. We agreed with the reviewer that it has long been known that LDLR undergoes ectodomain shedding, which was stated in the Introduction and Discussion of the manuscript. We also acknowledged the fact that LDLR shedding is mainly mediated by metalloproteinases and that LDLR shedding plays an important role in lipid metabolism. However, there are so many metalloproteinases. For example, ADAM proteins and matrix metalloproteinases each have more than 20 family members. **The question is which metalloproteinase is responsible for LDLR ectodomain shedding.** Considering the critical role of hepatic LDLR in the clearance of circulating LDL cholesterol, identification of the protein mediating LDLR shedding is very important since it will provide an alternative approach to lower plasma LDL cholesterol and the risk for cardiovascular disease. We have provided detailed evidence to prove that MT1-MMP is the metalloproteinase that promotes LDLR shedding in cultured human hepatocytes and mouse liver. The novel finding of our study is the identification of the metalloproteinase responsible for LDLR shedding. In addition, although it has not been documented in the literature that MT1-MMP can regulate plasma levels of LDL cholesterol, it has been reported that MT1-MMP affects the development of adipocytes. This indicates the potential regulatory role of MT1-MMP in lipid metabolism. Thus, we agree with the reviewer that the previous title indeed overemphasized the significance of our finding on the role of MT1-MMP in lipid metabolism. We have modified the title to more precisely represent our novel finding, MT1-MMP promoting LDLR shedding. The new title is “Membrane type 1 matrix metalloproteinase promotes ectodomain shedding of low-density lipoprotein receptor and accelerates the development of atherosclerosis”. In addition, Raw data of important experiments were provided in the revision.

Reviewer 2.

Alabi and coworkers report that membrane type 1-matrix metalloproteinase (MT1-MMP) promotes ectodomain cleavage of the low-density lipoprotein receptor and thus define a new role of membrane type 1 matrix metalloproteinase in lipid metabolism. Membrane type 1-matrix metalloproteinase (MT1-MMP) is a Zn²⁺-dependent endopeptidase that can cleave extracellular matrix and non-matrix substrates, such as LDLR. The major findings of this study include: 1) knockdown of MT1-MMP increased cellular LDLR abundance and reduced the levels of cleaved soluble ectodomain of LDLR (sLDLR) in cultured hepatocytes; 2) LDLR may interact with MT1-MMP, which is demonstrated by using co-immunoprecipitation and confocal microscopy; 3) mice lacking hepatic MT1-MMP displayed an increase in liver LDLR levels and a reduction in plasma levels of sLDLR, HDL-cholesterol, and non-HDL cholesterol; 4) the majority of sLDLR were associated with apolipoprotein B (apoB) and apoE-containing lipoproteins in both mouse and human plasma. The authors demonstrate that MT1-MMP promotes the ectodomain shedding of LDLR and plays an important role in lipid metabolism. These data implicate that MT1-MMP promotes the ectodomain shedding of LDLR and is crucial for lipid metabolism. It is an interesting study, however, there are several issues to be addressed.

Specific Comments:

1. The findings that MT1-MMP post-translationally mediates LDLR cleavage and regulates lipid metabolism are very interesting and potentially important. To enhance the significance of this study, it would be important to demonstrate the role of liver specific knockdown or overexpression of MT1-MMP in the development of atherosclerosis using a hypercholesterolemic mouse model.

Response. This is a great suggestion. We totally agree with the reviewer. We are currently breeding MT1-MMP liver specific knockout (MT1-MMP^{LKO}) mice with apoE knockout mice to knock out hepatic MT1-MMP in apoE knockout mice. Once we obtain the double knockout mice, we will examine the impact of lacking hepatic MT1-MMP on the development of atherosclerosis. However, this is a long procedure. It will take more than one and half years to obtain the double knockout mice. Meanwhile, we took the reviewer's suggestion to overexpress MT1-MMP in apoE^{-/-} mice. We made AAV-MT1-MMP and overexpressed MT1-MMP in apoE^{-/-} mice and found that overexpression of MT1-MMP significantly increased lesion area in apoE^{-/-} mice. Detailed results were presented in the revised manuscript (Lines 288 to 289 on page 13 and lines 290-296 on Page 14; Fig. 6j and supplementary Fig. 5j and k).

2. Fig 1a. Having a schematic of the original hypothesis above the data showing that the hypothesis was incorrect is something of a distraction. The schematic of the hypothesis could be eliminated as it is stated in the text. Alternatively, please separate the hypothesis from the results of Figure 1a by putting them in separate panels.

Response. We have deleted the schematic in Figure 1a.

3. There is a clear dose dependent relationship between MT1-MMP transfection and reduced expression of LDLR in Fig 1d. It would be important to determine the expression of PCSK9 in these samples.

Response. We have done the experiment and found that overexpression of MT1-MMP had no significant effect on expression of PCSK9 (The last sentence on Page 5 and the first sentence on Page 6 lines 105-106; Supplementary Figure 1c, lanes 4-6 vs. 1-3).

4. "Addition of PCSK9 reduced cellular LDLR levels in scrambled siRNA transfected cells, as well as in cells transfected with MT1-MMP siRNA (Fig. 1b). Thus, knockdown of MT1-MMP did not affect PCSK9-promoted LDLR degradation." It seems MT1-2 siRNA transfection indeed increased PCSK9 expression and increased LDLR without addition of PCSK9 in Figure 1b. But PCSK9 is a potent way to degrade the LDLR. It might be important to confirm the results with the co-transfection of PCSK9 and MT1-2 in this experiment rather than addition of exogenous PCSK9.

Response. We did the co-transfection experiment as the reviewer suggested and observed that knockdown of MT1-MMP had no significant effect on PCSK9-promoted LDLR degradation. The results were presented in the revised manuscript (Lines 93 to 96 on page 5; Supplementary Figure 1a)

5. The background of the Cal blot is too dark in Figure 1b. Please adjust the background to the same as others in Figure 1b.

Response. Sorry for that. We have adjusted the background of the blot as suggested.

6. Figure 5 d,e and f. The sample number is very small for a mouse study (n=3). The data would be strengthened by increasing the n. It would be important to quantify the TG levels and cholesterol levels in the livers in Figure 5f.

Response. We have quantified more samples in Figures 5d, e, and f and tested liver TG and cholesterol levels as suggested (Lines 226 to 228 on Page 11; Supplementary Figures 4a and b).

7. Please include the scale bar in Fig 3e, Fig 5e and Fig 5f.

Response. Sorry for missing the scale bar. It has been added to the figures in the revised manuscript.

8. Mice lacking hepatic MT1-MMP displayed an increase in liver LDLR levels and a reduction in plasma levels of sLDLR, HDL-cholesterol, and non-HDL cholesterol. It would be important to determine the expression levels of PCSK9 in these mice.

Response. This is a great suggestion. We have tested PCSK9 levels in plasma of MT1-MMP hepatic knockout mice and their wild type littermates. We found that lack of hepatic MT1-MMP had no significant effect on the levels of PCSK9 in plasma (Lines 203 to 205 on Page 10; Supplementary Figures 3c).

9. Specific knockout of MT1-MMP in mouse hepatocytes increased hepatic LDLR levels and reduced plasma levels of lipoprotein cholesterol. Although it has been shown LDLR levels were markedly increased in MT1-MMP knockdown cells in the absence of PCSK9 in human hepatocytes cell lines, does knockout of MT1-MMP affect the expression levels of PCSK9 in mouse hepatocytes?

Response. We have tested PCSK9 levels in the liver of MT1-MMP hepatic knockout mice and their wild type littermates. We found that lack of hepatic MT1-MMP had no significant effect on the levels of PCSK9 in hepatocytes (Lines 203 to 205 on Page 10; Supplementary Figures 3d).

10. In Figure 2, “(a) Inhibitors treatment. Huh7 cells transfected with empty pCDNA3.1 or HA-tagged MT1-MMP-pCDNA3.1 were incubated with MG132 (10 μ M, MG) or chloroquine (10 μ M, Chloro) for 6 h.” and “48 h later, whole cell lysate was prepared and applied to immunoblotting with antibodies as indicated. TFR, transferrin receptor.” Please use complete sentences in this Figure Legend and the others as well.

Response. We were very sorry for the mistakes. They were all corrected in the revision.

11. Figure 6. It is very interesting that the cholesterol levels are related to the soluble LDLR concentrations in humans. It would be important to measure PCSK9 levels in these samples. Is it possible to measure PCSK9 levels in these samples, or to obtain other samples that have been handled appropriately so the PCSK9 levels can be measured in the same samples with LDL-C and sLDLR? Please address this point in the discussion as well.

Response. We greatly appreciated this constructive suggestion. We measured plasma levels of PCSK9 in these patients and performed correlation analysis of PCSK9 and soluble LDLR, as well as PCSK9 and LDL cholesterol. There is no significant correlation between plasma PCSK9 and soluble LDLR in the whole population, women, or men. Similarly, plasma levels of PCSK9 did not significantly correlate to plasma LDL cholesterol in the whole group or in women, but did exhibit a positive association with plasma LDL cholesterol in men (Lines 279-285 on Page 13; Supplementary Figures 5 d to i). We also discussed the relevant findings in the Discussion (Lines 307-318 on page 15).

Minor comments.

Minor:

1. “sLDLR levels in culture medium (1:5 diluted in the Reagent Diluent) and plasma from each mouse (1:20 diluted in the Reagent Diluent) were assessed using the Human or Mouse LDLR DuoSet ELISA assay kit in accordance to manufacturer’s protocol.” Please include the kit information (company).

Response. The information has already been included in the revised manuscript.

2. Please include the scale bar of figure 3e, 5e and f in figures and describe in the figure legend.

Response. The scale bar has been added to Figures 3e and 5e.

3. The manuscript would benefit from careful editorial work. There are a number of distracting grammatical errors. (Page 5. To confirmed this finding,.. To confirm this finding,..)

Response. We were very sorry for the grammar errors in the manuscript. The manuscript has been edited by Donna Douglas, a native English speaker.

Reviewers' Comments:

Reviewer #1:

Remarks to the Author:

The article has been improved somewhat in the revision, but this reviewer is still concerned about the stellar level of this work. The authors rightfully state that there are many extracellular proteinases such as ADAMs and a large MMP family. It is stated that "The question is which [one?] metalloproteinase is responsible for LDLR ectodomain shedding". There might be more than one and there might be other extracellular proteinases that can cleave the receptor. The authors do not appreciate the complexity of extracellular proteolysis, but settle for a single MMP. This work really not clarify the role of MT1-MMP in vivo, but rather adds one more component to the confusion.

Reviewer #2:

Remarks to the Author:

The revised manuscript by Adekunle Alabi., et al seeks to investigate the effect of ectodomain cleavage of the low-density lipoprotein receptor and to define a new role of membrane type 1 matrix metalloproteinase in lipid metabolism. The authors have provided more evidence on the effect of AAV expression of human MT1-MMP cDNA on atherosclerosis in ApoE^{-/-} mice. However, there are several issues that need to be addressed with regard to the atherosclerosis experiment.

Specific Comments:

1. "We examined the effect of MT1-MMP on the development of atherosclerosis in mice. ApoE^{-/-} mice were injected with empty AAV or AAV containing human MT1-MMP cDNA and then fed the Western-type diet for 8 weeks".

With regard to the AAV-human MT1-MMP atherosclerosis study, the information regarding the methods used is not entirely clear (Fig. 6j and 6l Legends, Methods Page 32). Quantification of lesions in the proximal aorta are most often expressed as area: $\mu\text{m}^2 \times 10^3$, not % of aorta. In contrast, lesion area in en face preparation of the aortas is usually expressed as % of aorta. Variation in the lumen and folding make determining the % of aorta less reliable than calculating Oil-Red-O area. Did you make the calculation just based on one section (the sixth section) per mouse? This seems like a limited analysis, it is common to take the average of 5 -15 sections of the proximal aorta per mouse. It is not clear what this percentage represents. Is it the percentage of the aortic tissue area or the total cross sectional area including the lumen? It would be important to provide the units for the analysis for the proximal aorta as lesion area rather than relative percentage of aorta.

2. The brightness of Figure 6j is not homogeneous. Please replace them with better pictures.

3. "MT1-MMP knockdown mice displayed a trend of reduction in plasma levels of total cholesterol (mean value: 804 mg/dl in the control group vs. 659 mg/dl in MT1-MMP knockdown mice) (Fig. 6k), but the difference was not statistically significant".

What is the p value for the data in Fig 6k? The number of mice (n=6) used in Fig. 6k is relatively low, so it is not clear whether the lack of significant difference is real or due to limited power. It would be important to include more mice to have more robust results. Alternatively, this should be discussed as a limitation of the study.

4. It would be important to quantify the results from Supplemental Figure 6d to see if knockdown of MT1-MMP using AAV-TBG-Cre increased LDLR expression.

5. Supplemental Figure 6b. Was there a significant difference in the plasma cholesterol levels between groups? If so, please add the p value or indicate the difference with * in the figure and include the level of significance in the Legend.

6. Page 14. Lines 311- 313. Figure 6l and S63. Atherosclerosis experiment: "ApoE^{-/-} mice were injected with empty AAV or AAV-MT1-MMP (AAV-MT1) and then fed the Western-type diet for 8 weeks" . The presentation of the data is confusing. Are the sections shown in S6e supposed to be representative sections for the quantitation shown in Fig. 6l? Please provide the information on the gender and age of the mice used for the atherosclerosis analysis.

7. Figure 3a-3d. It looks like the IP blots of input and IP-beads were merged as the same picture. These should be run on the same western blot gel.

8. MT1-MMP activates pro-MMP2. The authors have previously reported that active MMP2 can cleave PCSK9 and a previous study has shown that MMP-2 inhibits PCSK9-induced degradation of the LDL receptor in Hepa1-c1c7 cells. In Figure 2b, in vitro knockdown of MT1-1 did not affect LDLR levels in the presence of PCSK9, but knockdown of MT1-1 increases LDLR expression in the absence of PCSK9 in Huh7. How do these results relate to the in vivo study (endogenous PCSK9) in Fig 6? Does MT1-MMP affect PCSK9-mediated LDLR degradation in vivo? This should be discussed.

Minor:

1. Please include the scale bar of supplemental figure 6e in figures and describe in the figure legend.
2. Donna N. Douglas c C. Her appointment is not defined on the title page.

Reviewer #1: The article has been improved somewhat in the revision, but this reviewer is still concerned about the stellar level of this work. The authors rightfully state that there are many extracellular proteinases such as ADAMs and a large MMP family. It is stated that “The question is which [one?] metalloproteinase is responsible for LDLR ectodomain shedding”. There might be more than one and there might be other extracellular proteinases that can cleave the receptor. The authors do not appreciate the complexity of extracellular proteolysis, but settle for a single MMP. This work really not clarify the role of MT1-MMP in vivo, but rather adds one more component to the confusion.

Response. As the reviewer mentioned, MT1-MMP can cleave a wide range of extracellular matrix components and non-matrix proteins and play an important role in many physiological and pathophysiological processes as we stated in the “Introduction”. However, compared to its roles in collagenolysis, our understanding of the non-extracellular matrix targets of MT1-MMP and their related physiological roles is poor. In this study, we identified a new substrate and a novel role of MT1-MMP, which is that MT1-MMP can regulate plasma levels of LDL cholesterol through mediating LDLR shedding. This finding further demonstrates the complexity of MT1-MMP’s physiological role. In addition, this study was mainly focused on identification of proteinase(s) responsible for LDLR shedding since LDLR, especially hepatic LDLR, is critical for the clearance of plasma LDL cholesterol. We found that knockout of hepatic MT1-MMP significantly reduced the levels of plasma soluble LDLR by approximately 67% (Fig. 4e). Considering the ubiquitous expression of LDLR, our finding indicates that, at least in the liver, MT1-MMP is the proteinase that contributes most to the ectodomain shedding of LDLR. We did not claim that MT1-MMP is the only proteinase that sheds the LDLR. As the reviewer mentioned, it is possible that other proteinases may also be involved in LDLR shedding. Therefore, We have added the following statements in the discussion.

“We noticed that knockout of hepatic MT1-MMP in mice markedly reduced the levels of plasma sLDLR approximately 67% and significantly increased liver LDLR levels (Figs. 4d and e). This indicates that, at least in the liver, MT1-MMP is the proteinase mainly responsible for the ectodomain shedding of LDLR. Given the ubiquitous expression of LDLR, LDLR shedding in other tissues may contribute to plasma sLDLR detected in *MT1*^{LKO} mice. However, we cannot rule out the possibility that other hepatic proteinases might also mediate LDLR shedding in the liver. For example, it has been reported that ADAM17 can promote LDLR shedding in HepG2 cells to a small extent¹⁵. More experiments are needed to define these possibilities.” (Page 17)

Reviewer#2:

The revised manuscript by Adekunle Alabi., et al seeks to investigate the effect of ectodomain cleavage of the low-density lipoprotein receptor and to define a new role of membrane type 1 matrix metalloproteinase in lipid metabolism. The authors have provided more evidence on the effect of AAV expression of human MT1-MMP cDNA on atherosclerosis in Apoe^{-/-} mice. However, there are several issues that need to be addressed with regard to the atherosclerosis experiment.

Specific Comments:

1. “We examined the effect of MT1-MMP on the development of atherosclerosis in mice. ApoE^{-/-} mice were injected with empty AAV or AAV containing human MT1-MMP cDNA and then fed the Western-type diet for 8 weeks”.

With regard to the AAV-human MT1-MMP atherosclerosis study, the information regarding the methods used is not entirely clear (Fig. 6j and 6l Legends, Methods Page 32). Quantification of lesions in the proximal aorta are most often expressed as area: $\mu\text{m}^2 \times 10^3$, not % of aorta. In contrast, lesion area in en face preparation of the aortas is usually expressed as % of aorta. Variation in the lumen and folding make determining the % of aorta less reliable than calculating Oil-Red-O area. Did you make the calculation just based on one section (the sixth section) per mouse? This seems like a limited analysis, it is common to take the average of 5 -15 sections of the proximal aorta per mouse. It is not clear what this percentage represents. Is it the percentage of the aortic tissue area or the total cross-sectional area including the lumen? It would be important to provide the units for the analysis for the proximal aorta as lesion area rather than relative percentage of aorta.

Response. The reviewer is correct. Thank you so much for the constructive comments. We collected 6 sections per mouse not just the sixth section for the analysis. We are sorry for the mistake. We have requalified the lesion area as the reviewer suggested. The method and data have been revised accordingly.

“We found that lesion area in the aortic sinuses was significantly increased in MT1-MMP-overexpressing mice ($157.5 \pm 21.03 \mu\text{m}^2 \times 10^3$ in the control group and $238.0 \pm 21.23 \mu\text{m}^2 \times 10^3$ in MT1-MMP overexpressing mice, $p=0.0228$; **Fig. 6j**).” (**Page 14**)
“and six sections per mouse were collected for the analysis” (**Page 33**)

2. The brightness of Figure 6j is not homogeneous. Please replace them with better pictures.

Response. The pictures in Figure 6j have been replaced with new ones with better qualities.

3. “MT1-MMP knockdown mice displayed a trend of reduction in plasma levels of total cholesterol (mean value: 804 mg/dl in the control group vs. 659 mg/dl in MT1-MMP knockdown mice) (Fig. 6k), but the difference was not statistically significant”.

What is the p value for the data in Fig 6k? The number of mice (n=6) used in Fig. 6k is relatively low, so it is not clear whether the lack of significant difference is real or due to limited power. It would be important to include more mice to have more robust results. Alternatively, this should be discussed as a limitation of the study.

Response. The p value is 0.2981. The reviewer is correct. Increasing the sample size tends to reduce the P value. However, the p value is not near the 0.05 cut-off point. Furthermore, the insignificant difference of plasma cholesterol between both groups may be because majority of plasma cholesterol in apoE^{-/-} mice is remnant cholesterol, which cannot be cleared by LDL

receptor due to the lack of apoE, the ligand of the LDL receptor. We have revised our statements in the “Results” and discussed the findings in “Discussion”.

“while hepatic LDLR levels were significantly increased in mice injected with AAV-TBG-Cre when compared to the AAV-GFP injected mice ($p=0.0385$, **Fig. S6d**). However, knockdown of hepatic MT1-MMP did not significantly affect plasma cholesterol levels ($p=0.2981$, Fig. 6k). Similarly, knockout of PCSK9 in apoE^{-/-} mice does not significantly affect plasma cholesterol levels or atherosclerotic plaque sizes despite increased hepatic LDLR levels²⁸.” (**Pages 14-15**)

“On the other hand, overexpression or knockdown of MT1-MMP in apoE^{-/-} mice did not significantly affect plasma cholesterol levels. Although we could not exclude the possibility that the insignificance might be caused by the relatively small sample size used in the study (six mice per group), it is of note that majority of plasma cholesterol in apoE^{-/-} mice is remnant cholesterol. apoE^{-/-} mice do not express apoE, the ligand of LDLR. Thus, chylomicron and VLDL remnants cannot be cleared by LDLR. This may explain why overexpression of MT1-MMP significantly increased and knockdown of MT1-MMP significantly reduced plasma cholesterol levels in mice with the wild-type background (Figs. 4g, 5b, and 5c) but not in apoE^{-/-} mice (Fig. 6k and Supple. Fig. 6b). Similarly, overexpression or knockout of PCSK9 does not significantly affect plasma cholesterol levels in apoE^{-/-} mice. Conversely, overexpression of PCSK9 enhances the development of atherosclerosis and knockout of PCSK9 reduces the levels of aortic cholesterol in apoE^{-/-} mice²⁸. Consistently, we found that the atherosclerotic lesion area in the aortic sinuses was significantly increased in apoE^{-/-} mice overexpressed MT1-MMP, while MT1-MMP knockdown reduced cholesteryl ester accumulated in the aorta of apoE^{-/-} mice.” (**Page 18**)

4. It would be important to quantify the results from Supplemental Figure 6d to see if knockdown of MT1-MMP using AAV-TBG-Cre increased LDLR expression.

Response. The quantification data has been presented in Supplementary Figure 6d.

“while hepatic LDLR levels were significantly increased in mice injected with AAV-TBG-Cre when compared to the AAV-GFP injected mice ($p=0.0385$, Fig. S6d).” (**Page 15**)

5. Supplemental Figure 6b. Was there a significant difference in the plasma cholesterol levels between groups? If so, please add the p value or indicate the difference with * in the figure and include the level of significance in the Legend.

Response. The difference is not statistically significant. $P=0.0772$. The p value was added to the text part. (**Page 14**).

“(730 mg/dL in the control, 911 mg/dL in MT1-MMP overexpressing mice, $p=0.0772$; **Fig. S6b**).” (**Page 14**)

We also discussed this finding in the revised manuscript. (**Page 18**)

“On the other hand, overexpression or knockdown of MT1-MMP in apoE^{-/-} mice did not significantly affect plasma cholesterol levels. Although we could not exclude the possibility that the insignificance might be caused by the relatively small sample size used in the study (six mice per group), it is of note that majority of plasma cholesterol in apoE^{-/-} mice is remnant cholesterol. apoE^{-/-} mice do not express apoE, the ligand of LDLR. Thus, chylomicron and VLDL remnants

cannot be cleared by LDLR. This may explain why overexpression of MT1-MMP significantly increased and knockdown of MT1-MMP significantly reduced plasma cholesterol levels in mice with the wild-type background (Figs. 4g, 5b, and 5c) but not in apoE^{-/-} mice (Fig. 6k and Supple. Fig. 6b). Similarly, overexpression or knockout of PCSK9 does not significantly affect plasma cholesterol levels in apoE^{-/-} mice. Conversely, overexpression of PCSK9 enhances the development of atherosclerosis and knockout of PCSK9 reduces the levels of aortic cholesterol in apoE^{-/-} mice²⁸. Consistently, we found that the atherosclerotic lesion area in the aortic sinuses was significantly increased in apoE^{-/-} mice overexpressed MT1-MMP, while MT1-MMP knockdown reduced cholesteryl ester accumulated in the aorta of apoE^{-/-} mice.” (Page 18)

6. Page 14. Lines 311- 313. Figure 6l and S63. Atherosclerosis experiment: "ApoE^{-/-} mice were injected with empty AAV or AAV-MT1-MMP (AAV-MT1) and then fed the Western-type diet for 8 weeks". The presentation of the data is confusing. Are the sections shown in S6e supposed to be representative sections for the quantitation shown in Fig. 6l? Please provide the information on the gender and age of the mice used for the atherosclerosis analysis.

Response. Sorry for the confusion. The reviewer is right. Images in Figure S6e were just representative sections. Both the representative images and quantification data have now been presented together as one figure (Supplementary Figure 6e) in the revision. Mice used in the atherosclerosis study were 8-week old male mice. The information was added in the method (Page 33) and the Legend to Figures 6j, k and l (Pages 57 and 59) and supplementary Figures 6a-e (Online Data Supplements, Pages 16 and 17).

7. Figure 3a-3d. It looks like the IP blots of input and IP-beads were merged as the same picture. These should be run on the same western blot gel.

Response. We have replaced the images in Figures 3a-d. Samples of Input and IP-Beads were run on the same Western blot gel in the new Figures. (Page 49).

8. MT1-MMP activates pro-MMP2. The authors have previously reported that active MMP2 can cleave PCSK9 and a previous study has shown that MMP-2 inhibits PCSK9-induced degradation of the LDL receptor in Hepa1-c1c7 cells. In Figure 2b, in vitro knockdown of MT1-1 did not affect LDLR levels in the presence of PCSK9, but knockdown of MT1-1 increases LDLR expression in the absence of PCSK9 in Huh7. How do these results relate to the in vivo study (endogenous PCSK9) in Fig 6? Does MT1-MMP affect PCSK9-mediated LDLR degradation in vivo? This should be discussed.

Response. Thank for the constructive comments. The pro- and active forms of MMP2 were not altered in MT1^{LKO} mice probably due to the action of extrahepatic MT1-MMP or other MMPs. Thus, it is unlikely that MMP2 makes a significant contribution to the action of MT1-MMP on LDLR. We added a statement in the revised manuscript.

“We have previously reported that active MMP2 inhibited PCSK9-promoted LDLR degradation in hepa1c1c7 cells²³. However, the levels of plasma pro- and active forms of MMP2 were not significantly altered in MT1^{LKO} mice (Fig. S3b), implying that MMP2 did not play an important role in the action of MT1-MMP on LDLR in mice.” (Page 10)

Figure 2b was the mRNA levels of different genes including *LDLR*. Knockdown of MT1-MMP did not affect mRNA levels of *LDLR*, indicating that the impact of MT1-MMP silencing on *LDLR* expression was on the posttranscriptional level. Data in Figure 1b indicated that knockdown of MT1-MMP appeared not to markedly affect the levels of *LDLR* in the presence of PCSK9 in Huh7 cells. This experiment was performed in the presence of excess PCSK9 and under a non-physiological condition, 1) Huh7 cells express endogenous PCSK9, 2) the cells were incubated with medium containing 5% lipoprotein-poor serum that is known to increase endogenous PCSK9 expression and enhance PCSK9-promoted *LDLR* degradation, and 3) we supplied the cells with additional 2 µg/ml of recombinant human PCSK9. Thus, it is likely that PCSK9-promoted *LDLR* degradation becomes overwhelming. Hence, we did not observe the impact of MT1-MMP silencing on *LDLR* expression. Experiments in Figure 6 represented the impact of MT1-MMP on *LDLR* under a normal physiological condition in the presence of a physiological concentration of PCSK9. We discussed these possibilities in the revision.

“We noticed that knockdown of MT1-MMP appeared not to markedly affect the levels of *LDLR* in the presence of exogenous PCSK9 (**Fig. 1b, lanes 4 and 6 vs 2**). It is of note that the experiment was performed in the presence of excess PCSK9 and under a non-physiological condition. First, Huh7 cells express endogenous PCSK9. Second, the cells were incubated in medium containing 5% NCLPPS that is known to increase endogenous PCSK9 expression and enhance PCSK9-promoted *LDLR* degradation. Third, the cells were supplied with additional 2 µg/ml of recombinant human PCSK9. Thus, it was likely that PCSK9-promoted *LDLR* degradation became overwhelming under this condition.” (**Pages 5-6**)

Minor:

1. Please include the scale bar of supplemental figure 6e in figures and describe in the figure legend.

Response. The scale bar has been added to the figure and figure legend. (**Online Data Supplements, Pages 16 and 17**)

2. Donna N. Douglas c C. Her appointment is not defined on the title page.

Response. Corrected.

Reviewers' Comments:

Reviewer #2:

Remarks to the Author:

The revised manuscript by Adekunle Alabi., et al seeks to investigate the effect of ectodomain cleavage of the low-density lipoprotein receptor and to define a new role of membrane type 1 matrix metalloproteinase in lipid metabolism. Some of the comments have been addressed adequately in the revision: the authors replaced relative atherosclerotic lesion data with the exact lesion area and added quantified results of hepatic LDLR levels. It is an interesting study, however, there are still some major issues to be addressed.

1. The authors have provided the gender information and the area of proximal aortic lesions in the revision. The lesion area of control apoe^{-/-} mice on WD for 8 weeks in Figure 6 is 8 times greater than what is reported for control apoe^{-/-} mice of the same gender after WD for 8 weeks in supplementary figure 6. This extraordinary difference suggests there is something wrong with the quantification or the description of the experiment. The revised manuscript by Adekunle Alabi., et al seeks to investigate the effect of ectodomain cleavage of the low-density lipoprotein receptor and to define a new role of membrane type 1 matrix metalloproteinase in lipid metabolism. Some of the comments have been addressed adequately in the revision: the authors replaced relative atherosclerotic lesion data with the exact lesion area and added quantified results of hepatic LDLR levels. It is an interesting study, however, there are still some major issues to be addressed.

1. The authors have provided the gender information and the area of proximal aortic lesions in the revision. The lesion area of control apoe^{-/-} mice on WD for 8 weeks in Figure 6 is 8 times greater than what is reported for control apoe^{-/-} mice of the same gender after WD for 8 weeks in supplementary figure 6. This extraordinary difference suggests there is something wrong with the quantification or the description of the experiment.

2. The brightness of Figure 6j is still not homogeneous.

3. Addressed

4. Addressed

5. "The difference is not statistically significant. $P=0.0772$. The p value was added to the text part. (Page 14). "(730 mg/dL in the control, 911 mg/dL in MT1-MMP overexpressing mice, $p=0.0772$; Fig. S6b). There is an obvious trend for an increase of plasma cholesterol levels in MT1-MMP overexpressing mice. Is it possible MT1-MMP affects the plasma cholesterol levels? Is this related to the increased hepatic LDLR expression. It would be important to increase the number of mice to evaluate whether this difference in plasma cholesterol levels is significant.

6. Supplementary Figure 6e, the variation of lesion area is so large that the comparison is meaningless. Also, it would be important to increase the number of mice or study more advanced lesions after 14-16 weeks on WD, which might reduce the variation.

7. Addressed.

8. Addressed.

Response

Reviewer #2 (Remarks to the Author):

The revised manuscript by Adekunle Alabi., et al seeks to investigate the effect of ectodomain cleavage of the low-density lipoprotein receptor and to define a new role of membrane type 1 matrix metalloproteinase in lipid metabolism. Some of the comments have been addressed adequately in the revision: the authors replaced relative atherosclerotic lesion data with the exact lesion area and added quantified results of hepatic LDLR levels. It is an interesting study, however, there are still some major issues to be addressed.

The authors have provided the gender information and the area of proximal aortic lesions in the revision. The lesion area of control *apoE*^{-/-} mice on WD for 8 weeks in Figure 6 is 8 times greater than what is reported for control *apoE*^{-/-} mice of the same gender after WD for 8 weeks in supplementary figure 6. This extraordinary difference suggests there is something wrong with the quantification or the description of the experiment. The revised manuscript by Adekunle Alabi., et al seeks to investigate the effect of ectodomain cleavage of the low-density lipoprotein receptor and to define a new role of membrane type 1 matrix metalloproteinase in lipid metabolism. Some of the comments have been addressed adequately in the revision: the authors replaced relative atherosclerotic lesion data with the exact lesion area and added quantified results of hepatic LDLR levels. It is an interesting study, however, there are still some major issues to be addressed.

1. The authors have provided the gender information and the area of proximal aortic lesions in the revision. The lesion area of control *apoE*^{-/-} mice on WD for 8 weeks in Figure 6 is 8 times greater than what is reported for control *apoE*^{-/-} mice of the same gender after WD for 8 weeks in supplementary figure 6. This extraordinary difference suggests there is something wrong with the quantification or the description of the experiment.

Response. Thank you for pointing out this issue. The two studies were done in two-month apart and the lesion areas were quantified by two different lab members. The overexpression experiment in Figure 6j was quantified by a more experienced postdoctoral fellow (PDF), while the knockdown experiment (Supplementary Figure 6e) was quantified by a graduate student. We went through the whole process together and found that the graduate student used a different setting in the microscope to quantify the lesion area. To determine if this caused the difference, the PDF who quantified the overexpressing experiment re-quantified the lesions of the knockdown experiment blindly. We also added 5 more mice to each group in the knockdown experiment as the reviewer recommended. A total number of mice in each group is 11 for Supplementary Figure 6e. The new mean lesion area of the knockdown experiment quantified by the PDF was $86.88 \mu\text{m}^2 \times 10^3$ in *apoE*^{-/-}/*MT1*^{Flox} mice. The previous data was $29.8 \mu\text{m}^2 \times 10^3$. Therefore, the different setting in the microscope used by the graduate student did cause a big difference in the quantification. We really appreciated the reviewer's comment so that we could find this big problem. In addition, we noticed that the new data of the knockdown experiment ($86.88 \mu\text{m}^2 \times 10^3$) was still less than that of the overexpression experiment ($157.7 \mu\text{m}^2 \times 10^3$). This difference might be caused by two reasons. **First**, *apoE*^{-/-} mice used in the MT1-MMP overexpressing study (Figure 6j) were purchased from the Jackson Laboratory and then housed in the animal facility at the University of Alberta during

the study. On the other hand, the mice used in the MT1-MMP knockdown experiment (Supplementary figure 6e) were apoE^{-/-}/MT1^{Flox} mice. MT1^{Flox} mice were backcrossed with C57BL/6J mice for more than 10 times and then bred with apoE^{-/-} mice purchased from the Jackson Laboratory to obtain apoE^{-/-}/MT1^{Flox} mice. apoE^{-/-}/MT1^{Flox} mice were housed in the same animal facility at the University of Alberta. It is possible that the difference in mouse background might cause this discrepancy. **Second**, the Western Diet used in the two studies were from different companies. The Diet used in the MT1-MMP overexpression study (Figure 6j) was from TestDiet (Catalogue # 1813029). Two months later, when we started the atherosclerosis study on the MT1-MMP knockdown mice (Supplementary figure 6e), the Western diet was purchased from Research Diets Inc (Catalogue # D12079Bi) due to a supply problem in TestDiet. Even though the two Western-type diets all contain 0.15% cholesterol and 40% energy from fat, the diet from the TestDiet uses milk fat, lard and soybean oil as the sources of fat and is a very soft pellet. The diet from Research Diets Inc. mainly uses anhydrous butter and corn oil as the fat sources and is a hard pellet. Thus, different foods might also contribute to the difference in atherosclerotic lesions observed in the two studies. We are very sorry for missing the information about the food and mice used in the two studies in the manuscript. They were included in the revised manuscript.

Although the experimental conditions were different between the overexpression (Figure 6j) and knockdown experiments (Supplementary figure 6e), we believe they did not affect our interpretation of the findings since the study was not to compare MT1-MMP overexpression with MT1-MMP knockdown. Instead, in the knockdown experiment, we compared apoE^{-/-}/MMP14^{Flox} mice injected with AAV-TBG-Cre (MT1-MMP knockdown) with apoE^{-/-}/MMP14^{Flox} mice injected with AAV-TBG-EGFP (the control). The mice had the same background (littermates) and were fed the same diet from Research Diets Inc. In the overexpression experiment, we compared apoE^{-/-} mice injected with AAV-TBG-MT1-MMP (MT1-MMP overexpression) with apoE^{-/-} mice injected with AAV-empty (the control). The mice used in this experiment also had the same background (apoE^{-/-} mice from the Jackson lab) and were fed the same diet from TestDiet. Thus, we are confident about our findings.

2. The brightness of Figure 6j is still not homogeneous.

Response. The image was replaced with a better one.

3. Addressed

4. Addressed

5. “The difference is not statistically significant. P=0.0772. The p value was added to the text part. (Page 14). “(730 mg/dL in the control, 911 mg/dL in MT1-MMP overexpressing mice, p=0.0772; Fig. S6b). There is an obvious trend for an increase of plasma cholesterol levels in MT1-MMP overexpressing mice. Is it possible MT1-MMP affects the plasma cholesterol levels? Is this related to the increased hepatic LDLR expression. It would be important to increase the number of mice to evaluate whether this difference in plasma cholesterol levels is significant.

Response. We increased the number of mice in each group to 13 and performed statistical analysis. The p value is 0.0283. Overexpression of MT1-MMP did significantly increase plasma cholesterol levels. The new data were shown in supplementary figure 6b.

6. Supplementary Figure 6e, the variation of lesion area is so large that the comparison is meaningless. Also, it would be important to increase the number of mice or study more advanced lesions after 14-16 weeks on WD, which might reduce the variation.

Response. We agreed with the reviewer and added more mice to each group. The number of mice in each group was 11. The new data were shown in Supplementary Figure 6e.

7. Addressed.

8. Addressed.

Reviewers' Comments:

Reviewer #2:

Remarks to the Author:

The authors have adequately addressed the issues raised and the manuscript is much improved. There are a couple of minor issues that should be addressed.

Minor

1. The lesion area appears much smaller in the current lesion pictures in Supplemental Figure 6e compared to the quantified results (86.88 $\mu\text{m}^2 \times 10^3$) if you compare it with the lesion area in Figure 6 (157.7 $\mu\text{m}^2 \times 10^3$). Can you provide more representative panels for Supplemental Figure 6e?

2. With regard to the description of Western Diet experiment (Page23), "For the high fat/high cholesterol experiment, mice were fed the Western- Type Diet containing 0.15% cholesterol from TestDiet or Research Diet Inc (kcal from fat 40%, protein 16%, and carbohydrate 44%)." This diet is not really a high cholesterol diet (the diet used in this study only has 0.15% cholesterol, the real high cholesterol diet usually has 1.25% cholesterol), so it is not accurate to say the high fat/high cholesterol experiment. It would be better to describe it as a high fat diet or Western diet experiment.

Response

REVIEWERS' COMMENTS

Reviewer #2 (Remarks to the Author):

The authors have adequately addressed the issues raised and the manuscript is much improved. There are a couple of minor issues that should be addressed.

Minor

1. The lesion area appears much smaller in the current lesion pictures in Supplemental Figure 6e compared to the quantified results (86.88 $\mu\text{m}^2 \times 103$) if you compare it with the lesion area in Figure 6 (157.7 $\mu\text{m}^2 \times 103$). Can you provide more representative panels for Supplemental Figure 6e?

Response. New images with more lesion area are used in the Supplemental Figure 6e.

2. With regard to the description of Western Diet experiment (Page23), “For the high fat/high cholesterol experiment, mice were fed the Western- Type Diet containing 0.15% cholesterol from TestDiet or Research Diet Inc (kcal from fat 40%, protein 16%, and carbohydrate 44%).” This diet is not really a high cholesterol diet (the diet used in this study only has 0.15% cholesterol, the real high cholesterol diet usually has 1.25% cholesterol), so it is not accurate to say the high fat/high cholesterol experiment. It would be better to describe it as a high fat diet or Western diet experiment.

Response. Sorry for the mistake. “the high fat/high cholesterol’ is changed to “the Western Diet” as suggested.